# Fast Proxy Experiment Design
# for Causal Effect Identification

**Sepehr Elahi**[*]
EPFL, Switzerland
sepehr.elahi@epfl.ch

**Sina Akbari**[*]
EPFL, Switzerland
sina.akbari@epfl.ch

**Jalal Etesami**
TUM, Germany
j.etesami@tum.de

**Negar Kiyavash**
EPFL, Switzerland
negar.kiyavash@epfl.ch

**Patrick Thiran**
EPFL, Switzerland
patrick.thiran@epfl.ch

## Abstract

Identifying causal effects is a key problem of interest across many disciplines. The two long-standing approaches to estimate causal effects are *observational* and *experimental (randomized)* studies. Observational studies can suffer from unmeasured confounding, which may render the causal effects unidentifiable. On the other hand, direct experiments on the target variable may be too costly or even infeasible to conduct. A middle ground between these two approaches is to estimate the causal effect of interest through *proxy experiments*, which are conducted on variables with a lower cost to intervene on compared to the main target. In an earlier work, we studied this setting and demonstrated that the problem of designing the optimal (minimum-cost) experiment for causal effect identification is NP-complete and provided a naive algorithm that may require solving exponentially many NP-hard problems as a sub-routine in the worst case. In this work, we provide a few reformulations of the problem that allow for designing significantly more efficient algorithms to solve it as witnessed by our extensive simulations. Additionally, we study the closely-related problem of designing experiments that enable us to identify a given effect through valid adjustments sets.

## 1 Introduction

Identifying *causal effects* is a central problem of interest across many fields, ranging from epidemiology all the way to economics and social sciences. While conducting randomized (controlled) trials provides a framework to analyze and estimate the causal effects of interest, such experiments are often impractical due to various limitations, including financial, logistical, and ethical constraints. Even when they are practical, gathering sufficient data to draw statistically significant conclusions is often challenging due to the high costs.

Costs can arise in multiple forms: financial costs (e.g., implementing costly interventions), time resources (e.g., upgrading infrastructure), and other

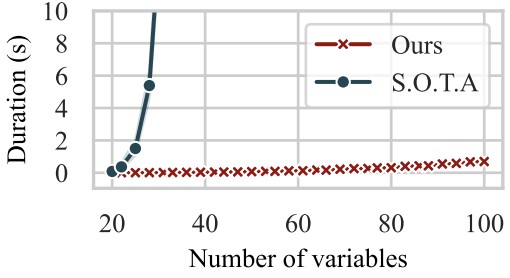

Figure 1: The average runtime of our approach compared with the state-of-the-art (S.O.T.A) from Akbari et al. [2022].

---

[*]Authors contributed equally.

38th Conference on Neural Information Processing Systems (NeurIPS 2024).

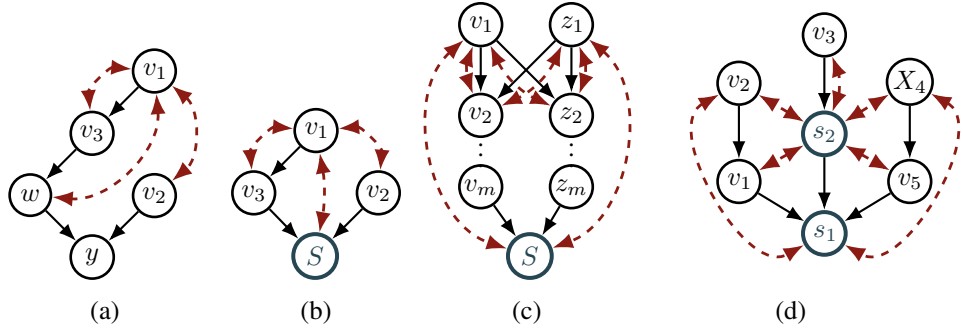

Figure 2: (a): Causal graph of Example 1. (b): Transformed graph when considering identifiability of $\mathbb{P}_{v_2,v_3}(y)$, with $S = \{w, y\}$. (c): A causal graph with $2^m$ minimal hedges. (d): Running example.

logistical constraints such as human resources. Moreover, certain experiments might be unethical or outright unfeasible, such as exposing patients to harmful treatments, and hence have infinite cost.

Observational data, which is usually more abundant and accessible, offers an alternative avenue. However, observational studies bring a new challenge: the causal effect may not be *identifiable* due to *unmeasured confounding*, making it impossible to draw inferences from the observed data [Pearl, 2009, Hernán and Robins, 2006]. A middle ground between observational and experimental approaches is to fuse data from both types of studies [Bareinboim and Pearl, 2016, Athey et al., 2020]. For example, cities might introduce low-emission zones as a proxy experiment before banning diesel vehicles [Dey et al., 2018], or governments might impose regional sugary drink taxes to estimate the effects of nationwide policies Redondo et al. [2018]. Several works have studied the achievability of identification from ensembles of observational and experimental data [Bareinboim and Pearl, 2012, Lee et al., 2020a, Kivva et al., 2022, Jamshidi et al., 2024]. Our previous work [Akbari et al., 2022] was the first to investigate the problem of designing *proxy experiments* to identify a causal effect that is not identifiable from observational data alone.

To illustrate the need for proxy experiments, consider the following drug-drug interaction example, based on the example in Lee et al. [2020a].

**Example 1.** *(Complex Drug Interactions and Cardiovascular Risk) Consider a simplified example involving the interaction between a new antihypertensive therapy ($v_1$), anti-diabetic medications ($v_2$), renal function modulators ($v_3$), and their effects on blood pressure ($w$) and cardiovascular disease ($y$). Blood pressure and cardiovascular health are closely linked. The antihypertensive therapy $v_1$ directly influences the need for renal function modulators $v_3$, and $v_3$ in turn directly affects blood pressure $w$. Additionally, anti-diabetic medications $v_2$ reduce the risk of cardiovascular disease $y$ by controlling blood sugar levels, while blood pressure $w$ directly impacts $y$. Unmeasured factors confound these relationships: shared health conditions can influence the prescription of both $v_1$ and $v_3$; lifestyle factors affect both $v_1$ and $w$; and common conditions like metabolic syndrome can impact both $v_1$ and $v_2$. Fig. 2(a) illustrates the resulting causal graph, whose directed edges represent direct causal effects, and bidirected edges indicate unmeasured confounders. Suppose we are interested in estimating the intervention effects of $v_2$ and $v_3$ on $y$, which are not identifiable from observational data alone. Moreover, we cannot directly intervene on these variables because $v_2$ and $v_3$ are essential for managing immediate, life-threatening conditions. Instead, we can intervene on $v_1$, which is a feasible and safer approach due to the broader range of treatment options and more manageable risks associated with adjusting antihypertensive therapy. As we shall see, intervention on $v_1$ suffices for identifying the effects of $v_2$ and $v_3$ on $y$.*

Selecting the *optimal* set of proxy experiments is not straightforward in general. In particular, in [Akbari et al., 2022] we proved that the problem of finding the *minimum-cost intervention set* to identify a given causal effect, hereon called the MCID problem, is NP-complete and provided a naive algorithm that requires solving exponentially many instances of the minimum hitting set problem in the worst case. As the minimum hitting set problem is NP-complete itself, our earlier algorithm in [Akbari et al., 2022] can become computationally intractable even for graphs with a modest number of vertices. Moreover, this algorithm was tailored to a specific class of causal effects in which the effect of interest is a functional of an interventional distribution where the intervention is made on

every variable except one *district* of the causal graph[2]. For a general causal effect, the complexity of this algorithm includes an additional (super-)exponential multiplicative factor, where the exponent is the number of districts.

In this work, we revisit the MCID problem and develop tractable algorithms by reformulating the problem as instances of well-known problems, such as the weighted maximum satisfiability and integer linear programming problems. Furthermore, we analyze the problem of designing minimum cost interventions to obtain a valid adjustment set for a query. This problem not only merits attention in its own right, but also serves as a proxy for MCID. Our contributions are as follows:

- We formulate the MCID problem in terms of a partially weighted maximum satisfiability, integer linear programming, submodular function maximization, and reinforcement learning problem. While our main focus is on the former two reformulations, we state the others in Appendix D.

- We propose new, and in practice, much faster (up to six orders of magnitude) algorithms for solving the problem optimally using our reformulations. Moreover, the computational complexity of our proposed algorithms scales quadratically in the number of districts, a significant improvement over the super-exponential growth exhibited by existing algorithms – see Remark 2.

- We formulate and study the problem of designing minimum-cost experiments for identifying a given effect through finding a valid adjustments set. Besides the practical advantages of valid adjustment, including ease of interpretability and tractable sample complexity, this approach enables us to design a polynomial-time heuristic algorithm for the MCID problem that outperforms the heuristic algorithms provided in Akbari et al. [2022].

- We present new numerical experiments that demonstrate the exceptional speed of our exact algorithms when compared to the current state-of-the-art, along with our heuristic algorithm showcasing superior performance over previous heuristic approaches.

## 2    Problem formulation

We begin by reviewing relevant graphical definitions. An *acyclic directed mixed graph* (ADMG) is a graph with directed ($\rightarrow$) and bidirected ($\leftrightarrow$) edges such that the directed edges form no cycles [Richardson, 2003]. We denote an ADMG $\mathcal{G}$ by a tuple $\mathcal{G} = \langle V, \overrightarrow{E}, \overleftrightarrow{E} \rangle$, where $V$, $\overrightarrow{E}$, and $\overleftrightarrow{E}$ represent the set of vertices, directed edges, and bidirected edges, respectively. Note that $\overrightarrow{E}$ is a set of ordered pairs of vertices in $V$, whereas $\overleftrightarrow{E}$ is a set of unordered pairs of vertices.

Vertices of $\mathcal{G}$ represent variables of the system under consideration, while the edges represent causal relations between them. We use the terms 'variable' and 'vertex' interchangeably. When $(y, x) \in \overrightarrow{E}$, we say $y$ is a parent of $x$ and $x$ is a child of $y$. The set of parents of $X \subseteq V$ denoted by $\mathrm{Pa}(X) = \{y : (y, x) \in \overrightarrow{E} \text{ for some } x \in X\} \setminus X$. We denote by $\mathcal{G}[W]$, the induced subgraph of $\mathcal{G}$ over vertices $W \subseteq V$. A subset $S$ of $V$ is said to form a *district* in $\mathcal{G}$ if $S$ is a maximal set such that any pair of vertices $x, y \in S$ are connected through a bidirected path $x \leftrightarrow \cdots \leftrightarrow y$ in $\mathcal{G}[S]$. In other words, $\mathcal{G}[S]$ is a connected component through its bidirected edges. We say $x \in V$ is an ancestor of $S \subseteq V$ if there is a directed path $x \rightarrow \cdots \rightarrow s$ for some $s \in S$. We denote the set of ancestors of $S$ in the subgraph $\mathcal{G}[W]$ by $\mathrm{Anc}_W(S)$. Note that $S \subseteq \mathrm{Anc}_W(S)$. When $W = V$, we drop the subscript for ease of notation.

Let $X, Y \subseteq V$ be two disjoint sets of variables. The probability distribution of $Y$ under a (possibly hypothetical) intervention on $X$ setting its value to x is often represented as either $\mathbb{P}(Y^{(\mathrm{x})})$, using Rubin's potential outcomes model [Rubin, 1974], or $\mathbb{P}(Y \mid \mathrm{do}(X = \mathrm{x}))$ using Pearl's do operator [Pearl, 2009]. We will adopt the shorthand $\mathbb{P}_\mathrm{x}(Y)$ to denote this interventional distribution[3].

**Definition 1** (Identifiability). *An interventional distribution $\mathbb{P}_\mathrm{x}(Y)$ is identifiable given an ADMG $\mathcal{G}$ and the intervention set family $\boldsymbol{\mathcal{I}} = \{\mathcal{I}_1, \ldots, \mathcal{I}_t\}$, with $\mathcal{I}_i \subseteq V$, over the variables corresponding to $\mathcal{G}$, if $\mathbb{P}_\mathrm{x}(Y)$ is uniquely computable as a functional of the members of $\{\mathbb{P}_\mathcal{I}(\cdot) : \mathcal{I} \in \boldsymbol{\mathcal{I}}\}$.*

---

[2]See Section 2 the definition of a district.

[3]This interventional distribution is often mistakenly referred to as the *causal effect* of $X$ on $Y$. However, a causal effect, such as an average treatment effect or a quantile treatment effect, is usually a specific functional of this probability distribution for different values of x.

**Remark 1.** *It is common in the literature to define identifiability with respect to observational data only (i.e., when $\mathcal{I} = \{\mathcal{I}_1 = \emptyset\}$). Our definition above follows what is known as the 'general identifiability' from Lee et al. [2020a].*

We will now formally define a *hedge*, which, as we will see shortly after, is central to deciding the identifiability of an interventional distribution given the data at hand.

**Definition 2** (Hedge). *Let $S \subseteq V$ be a district in $\mathcal{G}[S]$. We say $W \supsetneq S$ forms a hedge for $S$ if (i) $W$ is a district in $\mathcal{G}[W]$, and (ii) every vertex $w \in W$ is an ancestor of $S$ in $\mathcal{G}[W]$ (i.e., $W = \mathrm{Anc}_W(S)$). We denote by $H_{\mathcal{G}}(S)$ the set of hedges formed for $S$ in $\mathcal{G}$.*

For example, in Fig. 2(b), $S$ has two hedges given by $H_{\mathcal{G}}(S) = \{S \cup \{v_3, v_1\}, S \cup \{v_3, v_1, v_2\}\}$.

**Definition 3** (Hedge hull Akbari et al., 2022). *Let $S$ be a district in ADMG $\mathcal{G}$. Also let $H_{\mathcal{G}}(S)$ be the set of all hedges formed for $S$ in $\mathcal{G}$. The union of all hedges in $H_{\mathcal{G}}(S)$, denoted by $\mathcal{H}_{\mathcal{G}}(S) = \bigcup_{W \in H_{\mathcal{G}}(S)} W$, is said to be the hedge hull of $S$ in $\mathcal{G}$.*

For instance, in Fig. 2(d), the hedge hull of $S_1 = \{s_1\}$ is $\mathcal{H}_{\mathcal{G}}(S_1) = \{s_1, s_2, v_1, v_2, v_3, v_4, v_5\}$ and the hedge hull of $S_2 = \{s_2\}$ is $\mathcal{H}_{\mathcal{G}}(S_2) = \{s_2, v_3\}$. When a set $S$ consists of more than one district, we simply define the hedge hull of $S$ as the union of the hedge hulls of each district of $S$. The hedge hull of a set can be found through a series of at most $|V|$ depth-first-searches. In the latter example, the hedge hull of $S = \{s_1, s_2\}$ is $\mathcal{H}_{\mathcal{G}}(S) = \{s_1, s_2, v_1, v_2, v_3, v_4, v_5\}$. For the sake of completeness, we have included the algorithm for finding a hedge hull in Appendix B.1.

The following proposition from Lee et al. [2020a] and Kivva et al. [2022] establishes the graphical criterion for deciding the identifiability of a causal effect given a set family of interventions.

**Proposition 1.** *Let $\mathcal{G}$ be an ADMG over the vertices $V$. Also let $X, Y \subseteq V$ be disjoint sets of variables. Define $S = \mathrm{Anc}_{V \setminus X}(Y)$, and let $\boldsymbol{S} = \{S_1, \ldots, S_r\}$ be the (unique) set of districts in $\mathcal{G}[S]$. The interventional distribution $\mathbb{P}_x(Y)$ is identifiable given $\mathcal{G}$ and the intervention set family $\boldsymbol{\mathcal{I}} = \{\mathcal{I}_1, \ldots, \mathcal{I}_t\}$, if and only if for every $S_\ell \in \boldsymbol{S}$, there exists an intervention set $\mathcal{I}_k \in \boldsymbol{\mathcal{I}}$ such that (i) $\mathcal{I}_k \cap S_\ell = \emptyset$, and (ii) there is no hedge formed for $S_\ell$ in $\mathcal{G}[V \setminus \mathcal{I}_k]$.*

Note that there is no hedge formed for $S_\ell$ in $\mathcal{G}[V \setminus \mathcal{I}_k]$ if and only if $\mathcal{I}_k$ *hits* every hedge of $S_\ell$ (i.e., for any hedge $W \in \mathcal{H}_{\mathcal{G}}(S_\ell)$, $\mathcal{I}_k \cap W \neq \emptyset$). For ease of presentation, we will use $\mathcal{I}_k \xrightarrow{\mathrm{id}} S_\ell$ to denote that $\mathcal{I}_k \cap S_\ell = \emptyset$ and $\mathcal{I}_k$ hits every hedge formed for $S_\ell$. For example, given the graph in Fig. 2(d) and with $\boldsymbol{S} = \{S_1, S_2\}$, an intervention set family that hits every hedge is $\boldsymbol{\mathcal{I}} = \{\{s_2\}, \{v_3\}\}$.

**Minimum-cost intervention for causal effect identification (MCID) problem.** Let $C : V \to \mathbb{R}^{\geq 0} \cup \{+\infty\}$ be a known function[4] indicating the cost of intervening on each vertex $v \in V$. An infinite cost is assigned to variables where an intervention is not feasible. Given $\mathcal{G}$ and disjoint sets $X, Y \subseteq V$, our objective is to find a set family $\boldsymbol{\mathcal{I}}^*$ with minimum cost such that $\mathbb{P}_x(Y)$ is identifiable given $\boldsymbol{\mathcal{I}}^*$; that is, for every district $S_\ell$ of $S$, there exists $\mathcal{I}_k$ such that $\mathcal{I}_k \xrightarrow{\mathrm{id}} S_\ell$. Since every $\mathcal{I} \in \boldsymbol{\mathcal{I}}$ is a subset of $V$, the space of such set families is the power set of the power set of $V$.

To formalize the MCID problem, we first write the cost of a set family $\boldsymbol{\mathcal{I}}$ as $C(\boldsymbol{\mathcal{I}}) := \sum_{\mathcal{I} \in \boldsymbol{\mathcal{I}}} \sum_{v \in \mathcal{I}} C(v)$, where with a slight abuse of notation, we denoted the cost of $\mathcal{I}$ by $C(\mathcal{I})$. The MCID problem then can be formalized as follows.

$$\boldsymbol{\mathcal{I}}^* \in \underset{\boldsymbol{\mathcal{I}} \in 2^{2^V}}{\mathrm{argmin}}\, C(\boldsymbol{\mathcal{I}}) \quad \textbf{s.t.} \quad \forall S_\ell \in \boldsymbol{S} : (\exists \mathcal{I}_k \in \boldsymbol{\mathcal{I}} : \mathcal{I}_k \xrightarrow{\mathrm{id}} S_\ell), \tag{1}$$

where $\boldsymbol{S} = \{S_1, \ldots, S_r\}$ is the set of districts of $S = \mathrm{Anc}_{V \setminus X}(Y)$, and $2^{2^V}$ represents the power set of the power set of $V$. In the special case where $S$ comprises a single district, the MCID problem can be presented in a simpler way.

**Proposition 2** (Akbari et al., 2022). *If $S = \mathrm{Anc}_{V \setminus X}(Y)$ comprises a single district $\mathcal{S} = \{S_1 = S\}$, then the optimization in (1) is equivalent to the following optimization:*

$$\mathcal{I}^* \in \underset{\mathcal{I} \in 2^{V \setminus S}}{\mathrm{argmin}}\, C(\mathcal{I}) \quad \textbf{s.t.} \quad \forall W \in H_{\mathcal{G}}(S) : \mathcal{I} \cap W \neq \emptyset. \tag{2}$$

*That is, the problem reduces to finding the minimum-cost set that 'hits' every hedge formed for $S$.*

---

[4]Although it only makes sense to assign non-negative costs to interventions, adopting non-negative costs is without loss of generality. If certain intervention costs are negative, one can shift all the costs equally so that the most negative cost becomes zero. This constant shift would not affect the minimization problem in any way.

Recall example Example 1, we were interested in finding the least costly proxy experiment to identify the effect of $v_2$ and $v_3$ on $y$. By Proposition 2, this problem is equivalent to finding an intervention set with the least cost (i.e., a set of proxy experiments) that hits every hedge of $S = \{y, w\}$ in the transformed graph (Fig. 2(b)). If $\mathcal{C}(v_1) < \mathcal{C}(v_3)$, then the optimal solution would be $\mathcal{I}^* = \{v_1\}$.

In the remainder of the paper, we consider the problem of identification of $\mathbb{P}_X(Y)$ for a given pair $(X, Y)$, and with $S$ defined as $S = \mathrm{Anc}_{V \setminus X}(Y)$, unless otherwise stated. We will first consider the case where $S$ comprises a single district, and then generalize our findings to multiple districts.

# 3 Reformulations of the min-cost intervention problem

In the previous section, we delineated the MCID problem as a discrete optimization problem. This problem, cast as Eq. (1), necessitates search within a doubly exponential space, which is computationally intractable. Algorithm 2 of [Akbari et al., 2022] is an algorithm that conducts this search and eventually finds the optimal solution. However, even when $S$ comprises a single district, this algorithm requires, in the worst case, exponentially many calls to a subroutine which solves the NP-complete minimum hitting set problem on exponentially many input sets, hence resulting in a doubly exponential complexity. More specifically, our previous algorithm described in Akbari et al. [2022] attempts to find a set of *minimal* hedges, where minimal indicates a hedge that contains no other hedges, and solves the minimum hitting set problem on them. However, there can be exponentially many minimal hedges, as shown for example in Fig. 2(c). Letting $m = n/2$, then any set that contains one vertex from each level (i.e., directed distance from $S$) is a minimal hedge, of which there are $\mathcal{O}(2^{n/2})$.

Furthermore, the computational complexity of Algorithm 2 of Akbari et al. [2022] grows super-exponentially in the number of districts of $S$. This is due to the necessity of exhaustively enumerating every possible partitioning of these districts and executing their algorithm once for each partitioning.

In this section, we reformulate the MCID problem as a weighted partially maximum satisfiability (WPMAX-SAT) problem [Fu and Malik, 2006], as well as an integer linear programming (ILP) problem. We focus on the WPMAX-SAT and ILP reformulations due to their computational efficiency and practical applicability, but we provide alternative reformulations as a submodular maximization problem and a reinforcement learning problem in Appendix D for completeness. The advantage of the WPMAX-SAT and ILP formulations is two-fold: (i) compared to Algorithm 2 of [Akbari et al., 2022], we state the problem as a *single* instance of another problem for which a range of well-studied solvers exist, and (ii) these formulations allow us to propose algorithms with computational complexity that is quadratic in the number of districts of $S$. We will see how these advantages translate to drastic performance gains in Section 5.

## 3.1 Min-cost intervention as a WPMAX-SAT problem

We begin with constructing a 3-SAT formula $F$ that is satisfiable if and only if the given query $\mathbb{P}_X(Y)$ is identifiable. To this end, we define $m + 2$ variables $\{x_{i,j}\}_{j=0}^{m+1}$ for each vertex $v_i \in V$, where $m = |\mathcal{H}_\mathcal{G}(S) \setminus S|$ is the cardinality of the hedge hull of $S$, excluding $S$. Intuitively, $x_{i,j}$ is going to indicate whether or not vertex $v_i$ is reachable from $S$ after $j$ iterations of alternating depth-first-searches on directed and bidirected edges. This is in line with the workings of Algorithm 2 for finding the hedge hull of $S$. In particular, if a vertex $v_i$ is reachable after $m + 1$ iterations, that is, $x_{i,m+1} = 1$, then $v_i$ is a member of the hedge hull of $S$. The query of interest is identifiable if and only if $\mathcal{H}_\mathcal{G}(S) = S$, that is, the hedge hull of $S$ contains no other vertices. Therefore, we ensure that the formula $F$ is satisfiable if and only if $x_{i,m+1} = 0$ for every $v_i \notin S$. The formal procedure for constructing this formula is as follows.

**SAT Construction Procedure.** Suppose a causal ADMG $\mathcal{G} = \langle V, \overrightarrow{E}, \overleftrightarrow{E} \rangle$ and a set $S \subset V$ are given, where $S$ is a district in $\mathcal{G}$. Suppose $\mathcal{H}_\mathcal{G}(S) = \{v_1, \ldots, v_n\}$ is the hedge hull of $S$ in $\mathcal{G}$, where without loss of generality, $S = \{v_{m+1}, \ldots v_n\}$, and $\{v_1, \ldots v_m\} \cap S = \emptyset$. We will construct a corresponding boolean expression in conjunctive normal form (CNF) using variables $\{x_{i,j}\}$ for $i \in \{1, \ldots, m\}$ and $j \in \{0, \ldots, m + 1\}$. For ease of presentation, we also define $x_{i,j} = 1$ for all $i \in \{m + 1, \ldots, n\}, j \in \{0, \ldots, m + 1\}$. The construction is carried out in $m + 2$ steps, where in each step, we conjoin new clauses to the previous formula using 'and'. The procedure is as follows:

- For odd $j \in \{1, \ldots, m+1\}$, for each directed edge $(v_i, v_\ell) \in \overrightarrow{E}$, add $(\neg x_{i,j-1} \lor x_{i,j} \lor \neg x_{\ell,j})$ to $F$.

- For even $j \in \{1, \ldots, m+1\}$, for each bidirected edge $\{v_i, v_\ell\} \in \overleftrightarrow{E}$, add both clauses $(\neg x_{i,j-1} \lor x_{i,j} \lor \neg x_{\ell,j})$ and $(\neg x_{\ell,j-1} \lor x_{\ell,j} \lor \neg x_{i,j})$ to $F$.

- Finally, at step $m+2$, add clauses $\neg x_{i,m+1}$ to the expression $F$ for every $i \in \{1, \ldots, m\}$.

As an example, consider the graph of Figure 3 with $n = 3$ vertices. The hedge hull of $S = \{v_3\}$ is $\{v_1, v_2, v_3\}$, and $m = |\{v_1, v_2\}| = 2$. Following the SAT construction procedure outlined above, the construction is carried out in $m + 2 = 4$ steps. Our SAT expression will consist of 8 variables, $x_{i,j}$ for $i \in \{1, 2\}$ – corresponding to $v_1$ and $v_2$ – and $j \in \{0, 1, 2, 3\}$ – corresponding to the four steps of construction. Below, we explain each step.

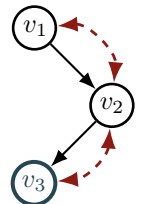

Figure 3: Example graph for 3SAT construction, where $S = \{v_3\}$.

At step 1, we add the clauses $(\neg x_{1,0} \lor x_{1,1} \lor \neg x_{2,1})$ and $(\neg x_{2,0} \lor x_{2,1} \lor \neg x_{3,1})$, corresponding to the edges $v_1 \to v_2$ and $v_2 \to v_3$, respectively. Note that by convention, $x_{3,1} = 1$, hence the second clause reduces to $(\neg x_{2,0} \lor x_{2,1})$. At step 2, for the edge $v_1 \leftrightarrow v_2$, we add the clauses $(\neg x_{1,1} \lor x_{1,2} \lor \neg x_{2,2})$ and $(\neg x_{2,1} \lor x_{2,2} \lor \neg x_{1,2})$. Similarly for the edge $v_1 \leftrightarrow v_3$, we add the clauses $(\neg x_{1,1} \lor x_{1,2} \lor \neg x_{3,2})$ and $(\neg x_{3,1} \lor x_{3,2} \lor \neg x_{1,2})$. Since by convention, $x_{3,1} = x_{3,2} = 1$, the latter two clauses reduce to $(\neg x_{1,1} \lor x_{1,2})$ and $1$, respectively. At step 3, we add the clauses $(\neg x_{1,2} \lor x_{1,3} \lor \neg x_{2,3})$ and $(\neg x_{2,2} \lor x_{2,3} \lor \neg x_{3,3})$, corresponding to the edges $v_1 \to v_2$ and $v_2 \to v_3$, respectively. Again $x_{3,3} = 1$, and the latter clause reduces to $(\neg x_{2,2} \lor x_{2,3})$. At step 4, the clauses $\neg x_{1,3}$ and $\neg x_{2,3}$ are added.

Finally, combining all the clauses together, the SAT expression is given by

$$(\neg x_{1,0} \lor x_{1,1} \lor \neg x_{2,1}) \land (\neg x_{2,0} \lor x_{2,1}) \land (\neg x_{1,1} \lor x_{1,2} \lor \neg x_{2,2}) \land (\neg x_{2,1} \lor x_{2,2} \lor \neg x_{1,2})$$
$$\land (\neg x_{1,1} \lor x_{1,2}) \land (\neg x_{1,2} \lor x_{1,3} \lor \neg x_{2,3}) \land (\neg x_{2,2} \lor x_{2,3}) \land \neg x_{1,3} \land \neg x_{2,3}.$$

**Theorem 1.** *The 3-SAT formula $F$ constructed by the procedure above given $\mathcal{G}$ and $S$ has a satisfying solution $\{x^*_{i,j}\}$ where $x^*_{i,0} = 0$ for $i \in \mathcal{I} \subseteq \{1, \ldots, m\}$ and $x^*_{i,0} = 1$ for $i \in \{1, \ldots, m\} \setminus \mathcal{I}$ if and only if $\mathcal{I}$ intersects every hedge formed for $S$ in $\mathcal{G}$; i.e., $\mathcal{I}$ is a feasible solution to the optimization in Eq. (2).*

The proofs of all our results appear in Appendix C. The first corollary of Theorem 1 is that the SAT formula is always satisfiable, for instance by setting $x^*_{i,0} = 0$ for every $i \in \{1, \ldots, m\}$. The second (and more important) corollary is that the optimal solution to Eq. (2) corresponds to the satisfying assignment for the SAT formula $F$ that minimizes

$$\sum_{i=1}^{m} (1 - x^*_{i,0}) C(v_i). \tag{3}$$

This suggests that the problem in Eq. (2) can be reformulated as a weighted partial MAX-SAT (WPMAX-SAT) problem. **WPMAX-SAT** is a generalization of the MAX-SAT problem, where the clauses are partitioned into *hard* and *soft* clauses, and each soft clause is assigned a weight. The goal is to maximize the aggregate weight of the satisfied soft clauses while satisfying all of the hard ones.

To construct the WPMAX-SAT instance, we simply define all clauses in $F$ as hard constraints, and add a soft clause $x_{i,0}$ with weight $C(v_i)$ for every $i \in \{1, \ldots, m\}$. The former ensures that the assignment corresponds to a feasible solution of Eq. (2), while the latter ensures that the objective in Eq. (3) is minimized – which, consequently, minimizes the cost of the corresponding intervention.

**Multiple districts.** The formulation above was presented for the case where $S$ is a single district. In the more general case where $S$ has multiple districts, we can extend our formulation to solve the general problem of Eq. (1) instead. To this end, we will use the following lemma.

**Lemma 1.** *Let $\mathcal{S} = \{S_1, \ldots, S_r\}$ be the set of districts of $S$, where $S = \mathrm{Anc}_{V \setminus X}(Y)$. There exists an intervention set family $\mathcal{I}^*$ of size $|\mathcal{S}| = r$ that is optimal for identifying $\mathbb{P}_X(Y)$.*

Based on Lemma 1, we can assume w.l.o.g. that the optimizer of Eq. (1) contains exactly $r$ intervention sets $\mathcal{I}_1, \ldots, \mathcal{I}_r$. We will modify the SAT construction procedure described in the previous section

to allow for multiple districts as follows. For any district $S_\ell$, we will construct $r$ copies of the SAT expression, one corresponding to each intervention set $\mathcal{I}_k$, $k \in \{1, \ldots, r\}$. Each copy is built on new sets of variables indexed by $(k, \ell)$, except the variables with index $j = 0$, which are common across districts. We introduce variables $\{z_{k,\ell}\}_{k,\ell=1}^r$, which will serve as indicators for whether $\mathcal{I}_k$ hits all the hedges formed for $S_\ell$. We relax every clause corresponding to the $k$-th copy by conjoining a $\neg z_{k,\ell}$ literal with an 'or.' Intuitively, this is because it suffices to hit the hedges formed for $S_\ell$ with some $\mathcal{I}_k$. Additionally, we add the clauses $(z_{1,\ell} \vee \cdots \vee z_{r,\ell})$ for any $\ell \in \{1, \ldots, r\}$ to ensure that for every district, there is at least one intervention set that hits every hedge. This modified procedure, detailed in Algorithm 3, appears in Appendix B.2. The following result generalizes Theorem 1.

**Theorem 2.** *Suppose $\mathcal{G}$, a set of its vertices $S$ with districts $\boldsymbol{S} = \{S_1, \ldots, S_r\}$, and an intervention set family 2 $\boldsymbol{\mathcal{I}} = \{\mathcal{I}_1, \ldots, \mathcal{I}_r\}$ are given. Define $m_\ell = |\mathcal{H}_\mathcal{G}(S_\ell) \setminus S_\ell|$, i.e., the cardinality of the hedge hull of $S_\ell$ excluding $S_\ell$ itself. The SAT formula $F$ constructed by Algorithm 3 has a satisfying solution $\{x_{i,0,k}^*\} \cup \{x_{i,j,k,\ell}^*\} \cup \{z_{k,\ell}^*\}$ where for every $\ell \in \{1, \ldots, r\}$, there exists $k \in \{1, \ldots, r\}$ such that (i) $z_{k,\ell}^* = 1$, (ii) $x_{i,0,k}^* = 0$ for every $i \in \mathcal{I}_k$, and (iii) $x_{i,0,k}^* = 1$ for every $i \in \{1, \ldots, m_\ell\} \setminus \mathcal{I}_k$, if and only if $\boldsymbol{\mathcal{I}}$ is a feasible solution to optimization of Eq. (1).*

Constructing the corresponding WPMAX-SAT instance follows the same steps as the case for a single district, except that the soft clauses are of the form $(x_{i,0,k} \vee \neg z_{k,\ell})$ with weight $C(v_i)$ for every $i \in \{1, \ldots, m_\ell\}$ and $k \in \{1, \ldots, r\}$.

**Remark 2.** *The SAT construction of Algorithm 3 is advantageous because its complexity grows quadratically with the number of districts of $S$ in the worst case. This is because the inner-loop of the SAT construction algorithm (line 8 of Algorithm 3) is executed $r^2$ many times. In contrast, the runtime of the algorithm of Akbari et al. [2022], when $S$ consists of multiple districts, is super-exponential in the number of districts, because they need to execute their single-district algorithm at least as many times as the number of partitions of the set $\{1, \ldots, r\}$.*

**Min-cost intervention as an ILP problem.** The WPMAX-SAT formulation of Section 3.1 paves the way for a straightforward formulation of an integer linear program (ILP) for the MCID problem. ILP allows for straightforward integration of various constraints and objectives, enabling flexible modeling of potential extra constraints. Moreover, there exist efficient and scalable solvers for ILP [Gearhart et al., 2013, Gurobi Optimization, LLC, 2023]. To construct the ILP instance for the MCID problem, it suffices to represent every clause in the boolean expression $F$ of Algorithm 3 as a linear inequality. For example, clauses of the form $(\neg a \vee b \vee \neg c)$ is rewritten as $(1 - a) + b + (1 - c) \geq 1$. The soft constraints may be rewritten as a sum to maximize over, given by Eq. (3).

# 4 Minimum-cost intervention design for adjustment criterion

A special case of identifying interventional distributions is identification through *adjusting* for confounders. A set $Z \subseteq V$ is a valid adjustment set for $\mathbb{P}_X(Y)$ if $\mathbb{P}_X(Y)$ is identified as

$$\mathbb{P}_X(Y) = \mathbb{E}_\mathbb{P}[\mathbb{P}(Y \mid X, Z)], \tag{4}$$

where the expectation w.r.t. $\mathbb{P}(Z)$. Adjustment sets have received extensive attention in the literature because of the straightforward form of the identification formula (Eq. 4) and the intuitive interpretation: $Z$ is the set of confounders that we need to *adjust for* to identify the effect of interest. The simple form of Eq. (4) has the added desirable property that its sample efficiency and asymptotic behavior are easy to analyze [Witte et al., 2020, Rotnitzky and Smucler, 2020, Henckel et al., 2022]. A complete graphical criterion for adjustment sets was given by Shpitser et al. [2010]. As an example, when all parents of $X$ (i.e., $\text{Pa}(X)$) are observable, they form a valid adjustment set. However, in the presence of unmeasured confounding, no valid adjustment sets may exist. Below, we generalize the notion of adjustment sets to the interventional setting.

**Definition 4** (Generalized adjustment). *We say $Z \subseteq V$ is a generalized adjustment set for $\mathbb{P}_X(Y)$ under intervention $\mathcal{I}$ if $\mathbb{P}_X(Y)$ is identified as $\mathbb{P}_X(Y) = \mathbb{E}_{\mathbb{P}_\mathcal{I}}[\mathbb{P}_\mathcal{I}(Y \mid X, Z)]$, where $\mathbb{P}_\mathcal{I}(\cdot)$ represents the distribution after intervening on $\mathcal{I} \subseteq V$ and the expectation is w.r.t. $\mathbb{P}_\mathcal{I}(Z)$.*

Note that unlike the classic adjustment, the generalized adjustment is always feasible – a trivial generalized adjustment can be formed by choosing $\mathcal{I} = X$ and $Z = \emptyset$.

Equipped with Definition 4, we can define a problem closely linked to Eq. (2), but with a (possibly) narrower set of solutions, which can be defined as follows: find the minimum-cost intervention $\mathcal{I}$ such that a generalized adjustment exists for $\mathbb{P}_X(Y)$ under $\mathcal{I}$:

$$\mathcal{I}^* = \underset{\mathcal{I} \in 2^V}{\arg\min} \, C(\mathcal{I}) \quad \textbf{s.t.} \quad \exists \, Z \subseteq V : \mathbb{P}_X(Y) = \mathbb{E}_{\mathbb{P}_\mathcal{I}}[\mathbb{P}_\mathcal{I}(Y \mid X, Z)]. \tag{5}$$

**Observation.** The existence of a valid (generalized) adjustment set ensures the identifiability of $\mathbb{P}_X(Y)$. As such, any feasible solution to the optimization above is also a feasible solution to Eq. (2). Eq. (5) is not only a problem that deserves attention in its own right, but also serves as a proxy for our initial problem (Eq. 2).

To proceed, we need the following definitions. Given an ADMG $\mathcal{G} = \langle V, \overrightarrow{E}, \overleftrightarrow{E} \rangle$, let $\mathcal{G}^d = \langle V^d, \overrightarrow{E}^d, \emptyset \rangle$ be the ADMG resulting from replacing every bidirected edge $e = \{x, y\} \in \overleftrightarrow{E}$ by a vertex $e$ and two directed edges $(e, x), (e, y)$. In particular, $V^d = V \cup \overleftrightarrow{E}$, and $\overrightarrow{E}^d = \overrightarrow{E} \cup \{(e, x) : e \in \overleftrightarrow{E}, x \in e\}$. Note that $\mathcal{G}^d$ is a directed acyclic graph (DAG). The *moralized graph* of $\mathcal{G}$, denoted by $\mathcal{G}^m$, is the undirected graph constructed by moralizing $\mathcal{G}^d$ as follows: The set of vertices of $\mathcal{G}^m$ is $V^d$. Each pair of vertices $x, y \in V^d$ are connected by an (undirected) edge if either (i) $(x, y) \in \overrightarrow{E}^d$, or (ii) $\exists z \in V^d$ such that $\{(x, z), (y, z)\} \subseteq \overrightarrow{E}^d$.

Throughout this section, we assume without loss of generality that $X$ is minimal in the following sense: there exists no proper subset $X_1 \subsetneq X$ such that $\mathbb{P}_X(Y) = \mathbb{P}_{X_1}(Y)$ everywhere[5]. Otherwise, we apply the third rule of do calculus [Pearl, 2009] as many times as possible to make $X$ minimal. We also assume w.l.o.g. that $V = \text{Anc}(X \cup Y)$ as other vertices are irrelevant for our purposes [Lee et al., 2020b]. We will utilize the following graphical criterion for generalized adjustment.

**Lemma 2.** *Let $X, Y$ be two disjoint sets of vertices in $\mathcal{G}$ such that $X$ is minimal as defined above. Set $Z \subseteq V$ is a generalized adjustment set for $\mathbb{P}_X(Y)$ under intervention $\mathcal{I}$ if (i) $Z \subseteq \text{Anc}(S)$, and (ii) $Z$ is a vertex cut[6] between $S$ and $\text{Pa}(S)$ in $(\mathcal{G}_{\overline{\mathcal{I}}\,\underline{\text{Pa}(S)}})^m$, where $S = \text{Anc}_{V \setminus X}(Y)$, and $\mathcal{G}_{\overline{\mathcal{I}}\,\underline{\text{Pa}(S)}}$ is the ADMG resulting from omitting all edges incoming to $\mathcal{I}$ and all edges outgoing of $\text{Pa}(S)$.*

Based on the graphical criterion of Lemma 2, we present the following *polynomial-time*[7] algorithm for finding an intervention set that allows for identification of the query of interest in the form of a (generalized) adjustment. This algorithm will find the intervention set $\mathcal{I}$ and the corresponding generalized adjustment set $Z$ simultaneously. We begin by making $X$ minimal in the sense of applicability of rule 3 of do calculus. Then we omit all edges going out of $\text{Pa}(S)$, and construct the graph $(\mathcal{G}_{\underline{\text{Pa}(S)}})^d = \langle V^d, \overrightarrow{E}^d, \emptyset \rangle$ as defined above – by replacing bidirected edges with vertices representing unobserved confounding. Finally, we construct an (undirected) vertex cut network $\mathcal{G}^{vc} = \langle V^{vc}, E^{vc} \rangle$ as follows. Each vertex $v \in V^d$ is represented by two connected vertices $v_1, v_2$ in $\mathcal{G}^{vc}$. If $v \in V$, then $v_1$ has a cost of zero, and $v_2$ has cost $C(v)$. Otherwise, both $v_1$ and $v_2$ have infinite costs. Intuitively, choosing $v_1$ will correspond to including $v$ in the adjustment set, whereas choosing $v_2$ in the cut would imply intervention on $v$. We connect $v_2$ to all vertices corresponding to $\text{Pa}(v)$ with index 1, i.e., $\{w_1 : (w, v) \in \overrightarrow{E}^d\}$. This serves two purposes: (i) if $v_2$ is included in the cut (corresponding to an intervention on $v$), all connections between $v$ and its parents are broken, and (ii) when $v_2$ is not included in the cut (corresponding to no intervention on $v$), $v_2$ connects the parents of $v$ to each other, completing the necessary moralization process. We solve for the minimum vertex cut between vertices with index 1 corresponding to $S$ and $\text{Pa}(S)$. Algorithm 1 summarizes this approach. In the solution set $J$, the vertices with index 2 represent the vertices where an intervention is required, while those with index 1 represent the generalized adjustment set under this intervention.

**Theorem 3.** *Let $(\mathcal{I}, Z)$ be the output returned by Algorithm 1 for the query $\mathbb{P}_X(Y)$. Then,*
*- $Z$ is a generalized adjustment set for $\mathbb{P}_X(Y)$ under intervention $\mathcal{I}$.*
*- $\mathcal{I}$ is the minimum-cost intervention for which there exists a generalized adjustment set based on the graphical criterion of Lemma 2.*

---

[5]This is to say, the third rule of do calculus does not apply to $\mathbb{P}_X(Y)$.

[6]This corresponds to $Z$ blocking all the backdoor paths between $\text{Pa}(S)$ and $S$, in the modified graph $\mathcal{G}_{\overline{\mathcal{I}}}$.

[7]The computational bottleneck of the algorithm is an instance of minimum vertex cut problem, which can be solved using any off-the-shelf max-flow algorithm.

---

**Algorithm 1** Intervention design for generalized adjustment

---

1: **procedure** MINCOSTGENADJUSTMENT$(X, Y, \mathcal{G} = \langle V, \overrightarrow{E}, \overleftrightarrow{E} \rangle, \{C(v) : v \in V\})$
2:     **while** $\exists x \in X$ s.t. $x$ is m-sep from $Y$ given $X \setminus \{x\}$ in $\mathcal{G}_{\overline{X}}$ **do**         $\triangleright$ Make $X$ minimal
3:         $X \leftarrow X \setminus \{x\}$
4:     $S \leftarrow \mathrm{Anc}_{V \setminus X}(Y)$
5:     $\overrightarrow{E} \leftarrow \overrightarrow{E} \setminus \{(x, s) : x \in \mathrm{Pa}(S)\}$         $\triangleright \mathcal{G}_{\mathrm{Pa}(S)}$
6:     $V^d \leftarrow V \cup \overrightarrow{E}, \quad \overrightarrow{E}^d \leftarrow \overrightarrow{E} \cup \{(e, v) : e \in \overleftrightarrow{E}, v \in e\}$         $\triangleright$ Construct $(\mathcal{G}_{\underline{\mathrm{Pa}(S)}})^d$
7:     $V^{vc} \leftarrow \cup_{v \in V^d} \{v_1, v_2\}, \quad E^{vc} \leftarrow \{\{v_1, v_2\} : v \in V^d\} \cup \{\{w_1, v_2\} : (w, v) \in \overrightarrow{E}^d\}$
8:     Construct a minimum vertex cut instance on the network $\mathcal{G}^{vc} = \langle V^{vc}, E^{vc} \rangle$, with costs 0 for any $v_1$ where $v \in V$, $C(v)$ for any $v_2$ where $v \in V$, and $\infty$ for any other vertex
9:     $J \leftarrow$ the minimum vertex cut between $\{x_1 : x \in \mathrm{Pa}(S)\}$ and $\{s_1 : s \in S\}$
10:     $\mathcal{I} \leftarrow \{v : v_2 \in J\}, \quad Z \leftarrow \{v : v_1 \in J\}$
11:     **return** $(\mathcal{I}, Z)$

---

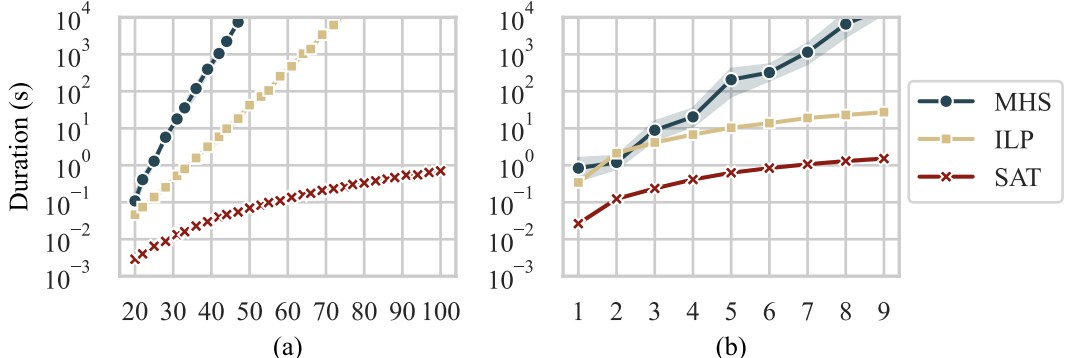

Figure 4: Average time taken by Algorithm 2 of Akbari et al. [2022] (MHS), ILP, and WPMAX-SAT to solve one graph versus (a) the number of vertices in the graph and (b) the number of districts of $S$.

**Remark 3.** *Algorithm 1 enforces identification based on (generalized) adjustment for $\mathbb{P}_X(Y)$. As discussed above, this algorithm can be utilized as a heuristic approach to solve the MCID problem in (2). In this case, one can run the algorithm on the hedge hull of $S$ rather than the whole graph. We prove in Appendix C that the cost of this approach is always at most as high as heuristic algorithm 1 of Akbari et al. [2022], and is often in practice lower, as verified by our experiments.*

The worst-case time complexity of Algorithm 1 is cubic in the number of variables. The two main computational bottlenecks are (i) the preprocessing in lines 2-3, which involves up to $|V|$ rounds of m-separation tests (performed via depth-first search), resulting in a complexity of $O\left(|V| \left[|V| + |\overrightarrow{E}| + |\overleftrightarrow{E}|\right]\right)$, and (ii) the minimum-cut instance in line 9, solved using a max-flow algorithm with $|V^{vc}| \sim O(|V| + |\overleftrightarrow{E}|)$ and $|E^{vc}| \sim O(|V| + |\overrightarrow{E}| + |\overleftrightarrow{E}|)$ many vertices and edges, respectively.

## 5 Experiments

In this section, we present numerical experiments that showcase the empirical performance and time efficiency of our proposed exact and heuristic algorithms. A comprehensive set of synthetic and real-world experiments analyzing the impact of various problem parameters on the performance of these algorithms, along with the complete implementation details, is provided in Appendix A. We first compare the time efficiency of our exact algorithms: WPMAX-SAT and ILP, with the exact algorithm of Akbari et al. [2022]. Then, we present results pertaining to performance of our heuristic algorithm. All experiments, coded in Python, were conducted on a machine equipped two Intel Xeon E5-2680 v3 CPUs, 256GB of RAM, and running Ubuntu 20.04.3 LTS.

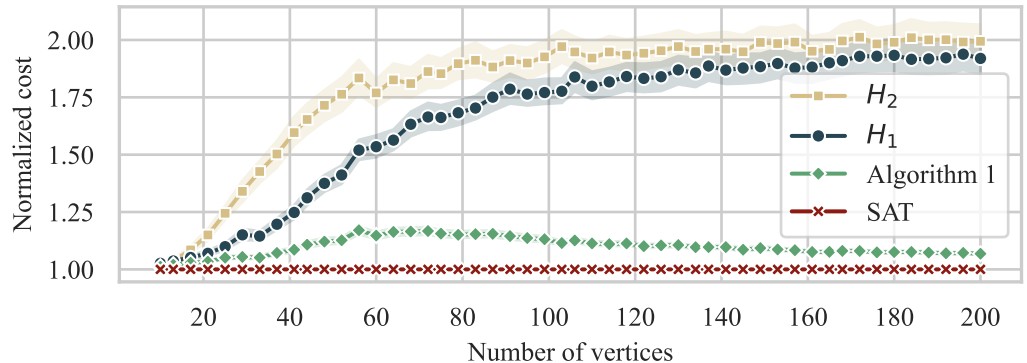

Figure 5: Average normalized cost of the heuristic algorithms $H_1$ and $H_2$ of Akbari et al. [2022] and Algorithm 1 versus the number of vertices in the graph.

**Results on exact algorithms.** We compare the performance of the WPMAX-SAT formulation, the ILP formulation, and Algorithm 2 of Akbari et al. [2022], called Minimal Hedge Solver (MHS) from hereon. We used the RC2 algorithm [Ignatiev et al., 2019], and the Gurobi solver [Gurobi Optimization, LLC, 2023], to solve the WPMAX-SAT problem, and the ILP, respectively. We ran each algorithm for solving the MCID problem on 100 randomly generated Erdos-Renyi [Erdos and Renyi, 1960] ADMG graphs with directed and bidirected edge probabilities ranging from 0.01 to 1.00, in increments of 0.01. We performed two sets of simulations: for single-district and multiple-district settings, respectively. In the single-district case, we varied $n$, the number of vertices, from 20 to 100, while in the multiple-district case, we fixed $n = 20$ and varied the number of districts from 1 to 9.

We plot the average time taken to solve each graph versus the number of vertices (single-district) in Fig. 4(a) and versus the number of districts ($n = 20$) in Fig. 4(b). The error bands in our figures represent 99% confidence intervals. Focusing on the single-district plot, we observe that both of our algorithms are faster than MHS of Akbari et al. [2022] for all graph sizes. More specifically, ILP is on average one to two orders of magnitude faster than MHS, while SAT is on average four to five orders of magnitude faster. All three algorithms exhibit exponential growth in time complexity with the number of vertices, which is expected as the problem is NP-hard, but SAT grows at a much slower rate than the other two algorithms. This is likely due to RC2's ability of exploiting the structure of the SAT problem to reduce the search space efficiently. In the multiple-district case, we observe that the time complexity of both SAT and ILP grows polynomially with the number of districts, while the time complexity of MHS grows exponentially. This is consistent with theory, as MHS iterates over all partitions of the set of districts, which grows exponentially with the number of districts.

**Results on inexact algorithms.** We compared Algorithm 1, our proposed heuristic, with the two best performing heuristic algorithms in Akbari et al. [2022], $H_1$ and $H_2$. We ran each algorithm on 500 randomly generated Erdos-Renyi ADMG graphs with directed and bidirected edge probabilities in $\{0.1, 0.5\}$, with $n$ ranging from 10 to 200. We randomly sampled the cost of each vertex from a discrete uniform distribution on $[1, n]$. In Fig. 5, we plot the normalized cost of each algorithm, computed by dividing the cost of the algorithm by the cost of the optimal solution, provided by WPMAX-SAT. Observe that Algorithm 1 consistently outperforms $H_1$ and $H_2$ for all graph sizes.

## 6 Conclusion

We introduced novel formulations and efficient algorithms for the MCID problem, demonstrating significant improvements over existing methods. Our extensive experiments showed that the WPMAX-SAT reformulation, particularly when using a high-performance solver like RC2, excels in both speed and effectiveness. In contrast, the ILP reformulation offers a more interpretable approach, especially valuable for incorporating additional constraints such as domain expert knowledge.

Moreover, our work on designing minimum-cost experiments for obtaining valid adjustment sets demonstrates both practical and theoretical advancements. We highlighted the superior performance of our proposed methods through extensive numerical experiments. We envision designing efficient approximation algorithms for MCID as future work.

## Acknowledgments

This research was in part supported by the Swiss National Science Foundation under NCCR Automation, grant agreement 51NF40_180545 and Swiss SNF projects 200021_182407 and 200021_204355 /1.

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

# Appendix

## A   Implementation details and further experimental results

### A.1   Implementation details

Our codebase is implemented fully in Python. We use the PySAT library for formulating and solving the WPMAX-SAT problem, and the PuLP library for formulating and solving the ILP problem.

**Solving the WPMAX-SAT problem.**   There are several algorithms to solve the WPMAX-SAT instance to optimality. These algorithms include RC2 [Ignatiev et al., 2019] and OLL [Morgado et al., 2014], both of which are core-based algorithms that utilize unsatisfiable cores to iteratively refine the solution. In this context, a "core" refers to an unsatisfiable subset of clauses within the CNF formula that cannot be satisfied simultaneously under any assignment. These algorithms relax the unsatisfiable soft clauses in the core by adding relaxation variables and enforce cardinality constraints on these variables. By strategically increasing the bounds on these cardinality constraints or modifying the weights of soft clauses based on the cores identified, the algorithms efficiently reduce the search space and converge on the maximum weighted set of satisfiable clauses, thereby solving the WPMAX-SAT problem optimally.

**Solving the ILP problem.**   Similarly, with the ILP formulation of the MCID problem presented in Section 3, we can utilize exact algorithms designed for solving ILP problems to find an optimal solution. ILP solvers work by formulating the problem with linear inequalities as constraints and integer variables that need to be optimized. Popular ILP solvers include CPLEX [IBM Corporation, 2023], Gurobi [Gurobi Optimization, LLC, 2023], and the open-source solver CBC [Forrest and Lougee-Heimer, 2023]. The latter is a branch-and-cut-based solver, and cutting plane methods to explore feasible integer solutions systematically while pruning the search space based on bounds calculated during the solving process.

We use the Gurobi solver in our experiments.

## A.2 Experiments on real-world data

We conduct experiments using 17 real-world networks from the Bayesian Network Repository[8]. This repository encompasses networks from diverse domains such as biology, engineering, medicine, and social science. In our experiments, each network from the repository is utilized as the DAG on the known variables. We assign a random cost to each variable, sampled uniformly at random from $[1, n]$. We then introduce hidden variables by randomly sampling bidirected edges with probabilities of 0.01, 0.1, and 0.3, resulting in 50 ADMGs per network. For each ADMG, we select $S$ to be a singleton consisting of the last node in the causal order. This choice ensures that any other node would not reduce the network's size by ignoring some vertices and edges, as outgoing edges from $S$ are irrelevant for the MCID problem.

Subsequently, we implemented the exact algorithms: minimal hedge solver (Algorithm 2 of Akbari et al. [2022]), ILP, and WPMAX-SAT. We also tested the heuristic algorithms $H_1$ and $H_2$ from Akbari et al. [2022], alongside Algorithm 1 from this work.

In Figure 6, we present a semi-log bar chart of the average time taken for each exact algorithm to solve an instance of a graph for each network. Notably, our approaches, ILP and SAT, consistently outperform MHS by an average factor of over 100, and by more than 1000 times on the largest network, link. Additionally, SAT demonstrates superior performance in all but two networks, *andes* and *diabetes*.

In Figure 7, we provide a semi-log plot of the normalized cost of the heuristics $H_1$ and $H_2$ from Akbari et al., 2022, and our Algorithm 1, averaged across each network. It is evident that Algorithm 1 significantly outperforms $H_1$, often by one to two orders of magnitude. Although $H_2$ performs better due to its focus on performing cuts on nodes over directed edges (which are fewer), its cost remains higher than that of Algorithm 1. Overall, the results align with our findings from synthetic simulations in the main text. Our WPMAX-SAT and ILP formulations surpass the previous state-of-the-art approach of Akbari et al. [2022], with WPMAX-SAT being the fastest. Furthermore, our polynomial-time Algorithm 1 consistently outperforms the heuristics of Akbari et al. [2022].

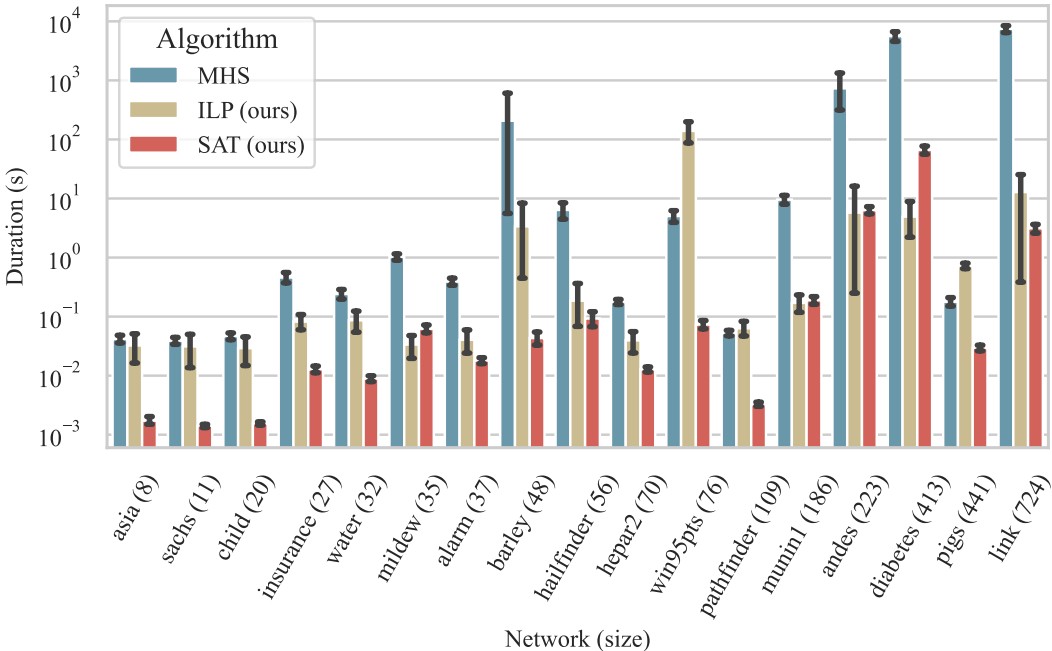

Figure 6: Semi-log plot of the average time taken by Algorithm 2 of Akbari et al. [2022] (MHS), ILP, and WPMAX-SAT to solve one instance of the network. Error bars represent the 99% confidence interval.

---

[8]bnlearn.com/bnrepository

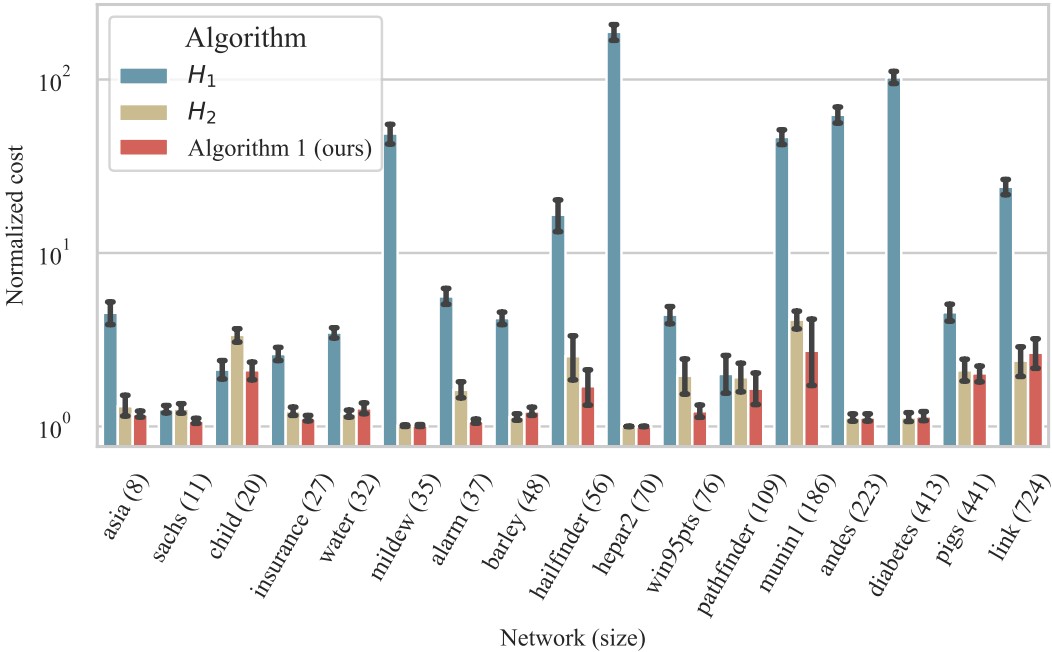

Figure 7: Semi-log plot of the average normalized cost of the heuristic algorithms $H_1$ and $H_2$ of Akbari et al. [2022] and Algorithm 1 to solve one instance of the network. Normalized cost is the cost of the solution divided by the cost of the optimal solution. Error bars represent the 99% confidence interval.

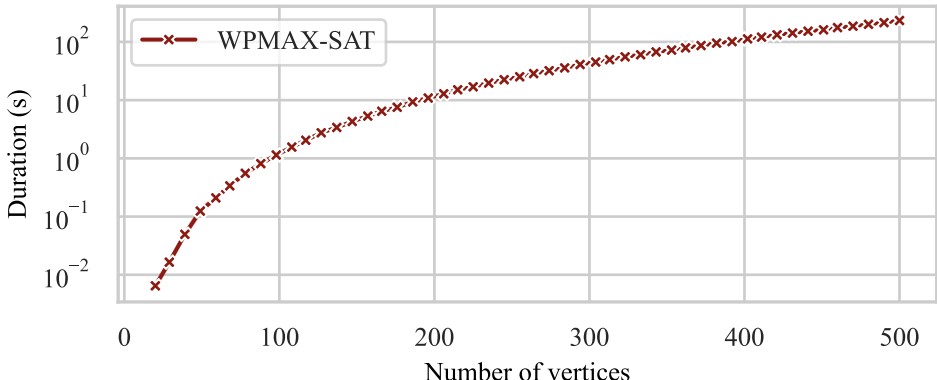

Figure 8: Semi-log plot of the average time taken by WPMAX-SAT to solve one graph versus the number of vertices in the graph.

### A.3 Extended WPMAX-SAT simulations

We extended the simulations in Section 5 for up to $n = 500$ vertices, and the results are presented in Fig. 8. We observe that even at $n = 500$, WPMAX-SAT takes around the same time as Algorithm 2 of Akbari et al. [2022] does to solve $n = 40$ (230 s for both). Moreover, we can clearly see the exponential growth in time complexity, as expected, especially for $n > 400$.

### A.4 Investigating the effects of directed and bidirected edge probabilities on the performance of exact algorithms

We run experiments on varying the probabilities of directed and bidirected edges in the graph. We fix the number of vertices at $n = 20$ and vary the probabilities of directed and bidirected edges from 0.001 to 1.00 in increments of 0.001. The results are presented in Fig. 9.

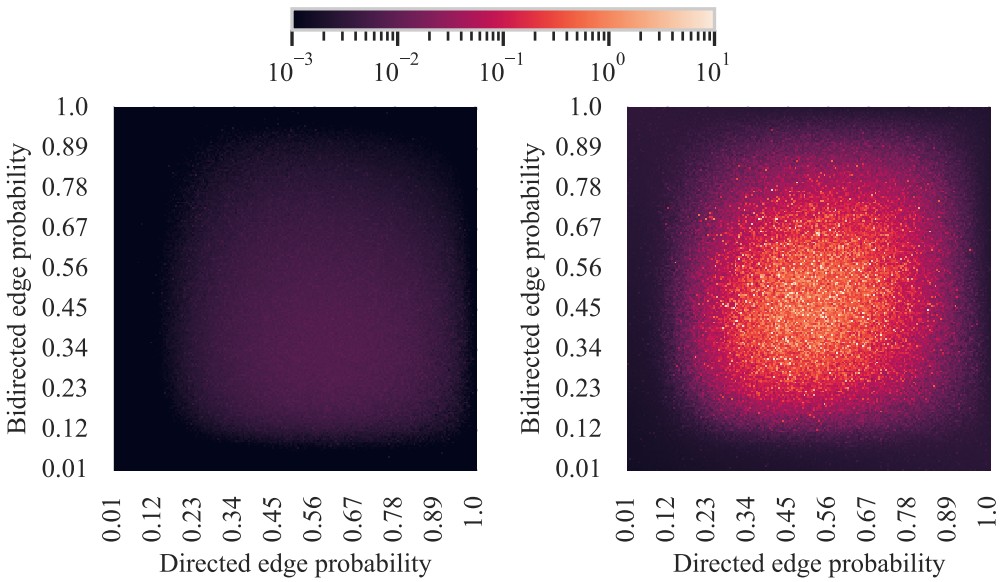

Figure 9: Heatmap of the average time taken by WPMAX-SAT (on the left) and Algorithm 2 of Akbari et al. [2022] (on the right) to solve one graph versus the probabilities of directed and bidirected edges in the graph.

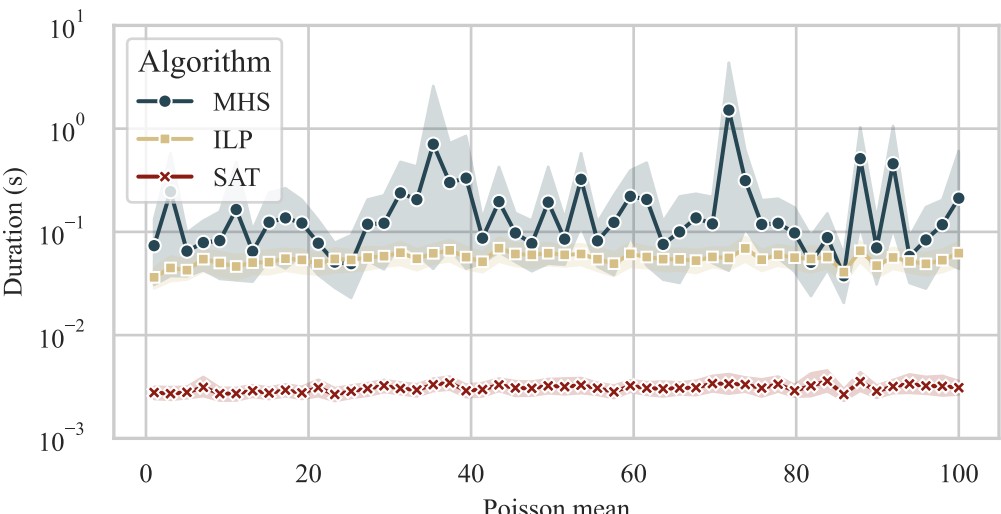

Figure 10: Average time taken by Algorithm 2 of Akbari et al. [2022] (MHS), ILP, and WPMAX-SAT to solve one graph versus the mean parameter of the Poisson distribution from which the costs are sampled.

## A.5 Investigating the effect of cost on the performance of the algorithms

We run experiments with $n = 20$ and costs sampled from a Poisson distributions with mean parameter ranging from 1 to 100. The results are presented in Fig. 10. Interestingly, there appears to be no clear trend in the time complexity of the algorithms with respect to the mean parameter of the Poisson distribution. This suggests that the time complexity of the algorithms is not significantly affected by the cost of the vertices.

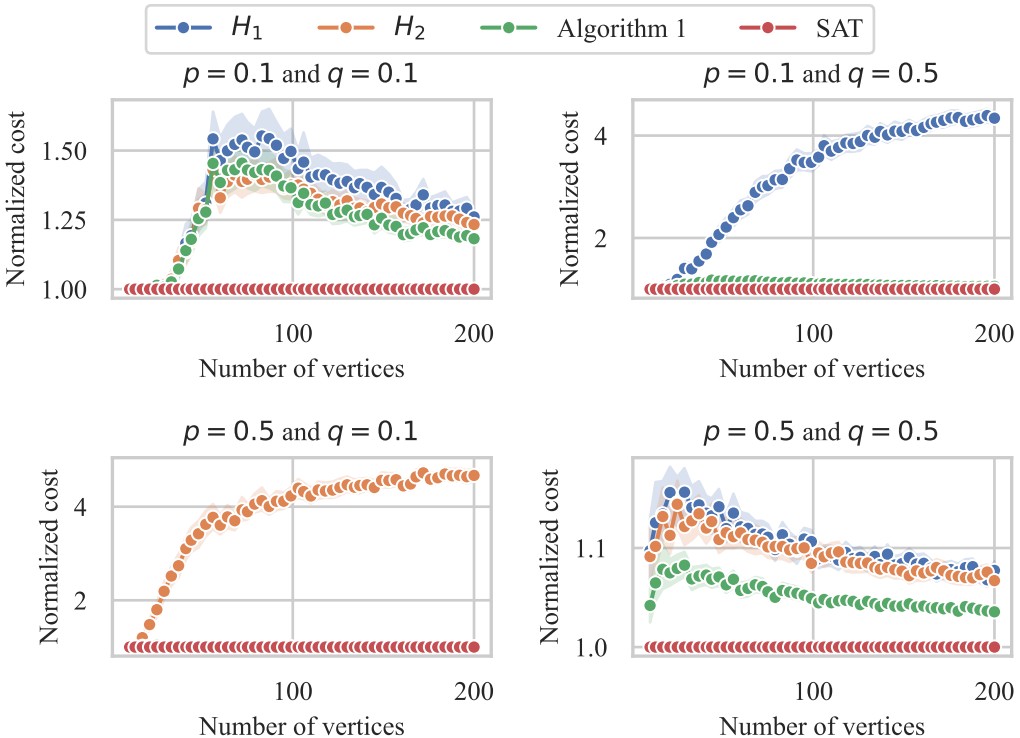

Figure 11: Average running time of the heuristic algorithms $H_1$ and $H_2$ of Akbari et al. [2022] and Algorithm 1 versus the number of vertices in the graph for different probabilities of directed and bidirected edges in the graph.

## A.6 Investigating the effects of directed and bidirected edge probabilities on the performance of the heuristic algorithms

We run experiments on varying the probabilities of directed and bidirected edges in the graph. We vary $n$ from $n = 10$ to $n = 200$ and the probabilities of directed and bidirected edges in $\{0.1, 0.5\}$. The results are presented in Fig. 11. We see that our proposed heuristic algorithm consistently outperforms the heuristic algorithms of Akbari et al. [2022] for all graph sizes and edge probabilities.

# B  Algorithms

## B.1  Pruning algorithm for finding the hedge hull

We include the algorithm for finding the hedge hull for the sake of completeness. This algorithm is adopted from Akbari et al. [2022].

---

**Algorithm 2** Pruning algorithm

---

1: **procedure** PRUNE($\mathcal{G} = \langle V, \overrightarrow{E}, \overleftrightarrow{E} \rangle, S$)
2:     $\mathcal{H} \leftarrow \text{Anc}_V(S)$
3:     **while** True **do**
4:         $\mathcal{H}' \leftarrow \{v \in \mathcal{H} : v \text{ has a bidirected path to } S \text{ in } \mathcal{G}[\mathcal{H}]\}$
5:         **if** $\mathcal{H}' = \mathcal{H}$ **then**
6:             **return** $\mathcal{H}$
7:         $\mathcal{H} \leftarrow \text{Anc}_{\mathcal{H}'}(S)$
8:         **if** $\mathcal{H} = \mathcal{H}'$ **then**
9:             **return** $\mathcal{H}$

---

## B.2  SAT construction procedure for multiple districts

The procedure for constructing the SAT formula when $S$ comprises multiple districts was postponed to this section due to space limitations. This procedure is detailed below.

---

**Algorithm 3** SAT construction

---

1: **procedure** CONSTRUCTSAT($X, Y, \mathcal{G} = \langle V, \overrightarrow{E}, \overleftrightarrow{E} \rangle$)
2:     $S \leftarrow \text{Anc}_{V \setminus X}(Y)$
3:     $\mathcal{S} \leftarrow$ districts of $S$ in $\mathcal{G}[S]$
4:     $r \leftarrow |\mathcal{S}|$
5:     $F \leftarrow 1$
6:     **for** $\ell \in \{1, \ldots r\}$ **do**              $\triangleright$ iterate over districts of $S$
7:         $m \leftarrow |\mathcal{H}_\mathcal{G}(S_\ell) \setminus S_\ell|$             $\triangleright$ # iterations
8:         **for** $k \in \{1, \ldots, r\}$ **do**         $\triangleright$ iterate over expressions
9:             $F \leftarrow F \wedge (x_{i,0,k} \vee \neg z_{k,\ell})$ for every $i$ s.t. $v_i \in S_\ell$
10:            $F \leftarrow F \wedge (x_{i,j,k,\ell} \vee \neg z_{k,\ell})$ for every $i$ s.t. $v_i \in S_\ell$ and every $j \in \{1, \ldots, m+1\}$
11:            **for** $(v_i, v_p) \in \overrightarrow{E}$ **do**         $\triangleright$ iteration $j = 1$
12:               $F \leftarrow F \wedge (\neg x_{i,0,k} \vee x_{i,1,k,\ell} \vee \neg x_{p,1,k,\ell} \vee \neg z_{k,\ell})$
13:            **for** $j \in \{2, \ldots, m+1\}$ **do**
14:               **if** $j$ is odd **then**
15:                  **for** $(v_i, v_p) \in \overrightarrow{E}$ **do**
16:                     $F \leftarrow F \wedge (\neg x_{i,j-1,k,\ell} \vee x_{i,j,k,\ell} \vee \neg x_{p,j,k,\ell} \vee \neg z_{k,\ell})$
17:               **else**
18:                  **for** $\{v_i, v_p\} \in \overleftrightarrow{E}$ **do**
19:                     $F \leftarrow F \wedge (\neg x_{i,j-1,k,\ell} \vee x_{i,j,k,\ell} \vee \neg x_{p,j,k,\ell} \vee \neg z_{k,\ell})$
20:                     $F \leftarrow F \wedge (\neg x_{p,j-1,k,\ell} \vee x_{p,j,k,\ell} \vee \neg x_{i,j,k,\ell} \vee \neg z_{k,\ell})$
21:            $F \leftarrow F \wedge (\neg x_{i,m+1,k,\ell} \vee \neg z_{k,\ell})$ for every $v_i \notin S_\ell$
22:         $F \leftarrow F \wedge (z_{1,\ell} \vee \cdots \vee z_{r,\ell})$
23:     **return** $F$

---

## C  Missing Proofs

### C.1  Results of Section 3

**Theorem 1.** *The 3-SAT formula $F$ constructed by the procedure above given $\mathcal{G}$ and $S$ has a satisfying solution $\{x_{i,j}^*\}$ where $x_{i,0}^*=0$ for $i \in \mathcal{I} \subseteq \{1,\ldots,m\}$ and $x_{i,0}^*=1$ for $i \in \{1,\ldots,m\} \setminus \mathcal{I}$ if and only if $\mathcal{I}$ intersects every hedge formed for $S$ in $\mathcal{G}$; i.e., $\mathcal{I}$ is a feasible solution to the optimization in Eq. (2).*

*Proof.  Proof of 'if:'* Suppose $\mathcal{I}$ hits every hedge formed for $S$. We construct a satisfying solution for the SAT formula as follows. We begin with $x_{i,0}$:

$$x_{i,0}^* = \begin{cases} 0; & \text{if } i \in \mathcal{I} \\ 1; & \text{o.w.} \end{cases}$$

For every $j \in \{1,\ldots,m+1\}$, define $H_j = \{i : x_{i,j-1} = 1\}$. Then $x_{i,j}^*$ for $j \in \{1,\ldots,m+1\}$ is chosen recursively as below.

- Odd $j$: $x_{i,j}^* = 1$ if $i \in H_j$ and $v_i$ has a directed path to $S$ in $\mathcal{G}[H_j]$, and $x_{i,j}^* = 0$ otherwise.

- Even $j$: $x_{i,j}^* = 1$ if $i \in H_j$ and $v_i$ has a bidirected path to $S$ in $\mathcal{G}[H_j]$, and $x_{i,j}^* = 0$ otherwise.

Next, we prove that $\{x_{i,j}^*\}$ as defined above satisfies $F$. We consider the three types of clauses in $F$ separately:

- For odd $j \in \{1,\ldots,m+1\}$, the clause $(\neg x_{i,j-1} \vee x_{i,j} \vee \neg x_{\ell,j})$ corresponds to the directed edge $(v_i, v_\ell) \in \overrightarrow{E}$: if either $x_{i,j-1}^* = 0$ or $x_{\ell,j}^* = 0$, then this clause is trivially satisfied. So suppose $x_{i,j-1}^* = 1$, and $x_{\ell,j}^* = 1$, which implies by construction that $x_{\ell,j-1}^* = 1$. Therefore, $i, \ell \in H_j$. Further, since $x_{\ell,j}^* = 1$, $v_\ell$ has a directed path to $S$ in $\mathcal{G}[H_j]$. Then $v_i$ has a directed path to $S$ in $\mathcal{G}[H_j]$ because of the edge $(v_i, v_\ell) \in \overrightarrow{E}$. By the construction above, $x_{i,j}^* = 1$, which satisfies the clause.

- For even $j \in \{1,\ldots,m+1\}$, the clause $(\neg x_{i,j-1} \vee x_{i,j} \vee \neg x_{\ell,j})$ corresponds to the bidirected edge $\{v_i, v_\ell\} \in \overleftrightarrow{E}$: if either $x_{i,j-1}^* = 0$ or $x_{\ell,j}^* = 0$, then this clause is trivially satisfied. So suppose $x_{i,j-1}^* = 1$, and $x_{\ell,j}^* = 1$, which implies by construction that $x_{\ell,j-1}^* = 1$. Therefore, $i, \ell \in H_j$. Further, since $x_{\ell,j}^* = 1$, $v_\ell$ has a bidirected path to $S$ in $\mathcal{G}[H_j]$. Then $v_i$ has a bidirected path to $S$ in $\mathcal{G}[H_j]$ because of the edge $\{v_i, v_\ell\} \in \overleftrightarrow{E}$. By the construction above, $x_{i,j}^* = 1$, which satisfies the clause.

- The clauses $\neg x_{i,m+1}$: First note that by construction, if for some $j \in \{1,\ldots,m+1\}$, $x_{i,j-1}^* = 0$, then $x_{i,j}^* = x_{i,j+1}^* = \cdots = x_{i,m}^* = 0$. That is, $\{x_{i,j}^*\}_{j=0}^m$ is a non-increasing binary-valued sequence for every $i$. Therefore, for every $i \in \{1,\ldots,m\}$, there exists *at most* one $j$ such that $x_{i,j-1}^* > x_{i,j}^*$. We consider two cases separately:

  - There are exactly $m$ many $j \in \{1,\ldots,m+1\}$ for which there exists at least one $i$ such that $x_{i,j-1}^* > x_{i,j}^*$. In this case, for every $i \in \{1,\ldots,m\}$, there exists exactly one $j \in \{1,\ldots,m+1\}$ such that $x_{i,j-1}^* > x_{i,j}^*$. Then for every $i$, there exists $j$ such that $x_{i,j}^* = 0$, and following the argument above, $x_{i,m+1}^* = 0$. Hence, the clauses $\neg x_{i,m+1}$ are all satisfied.

  - There are strictly less than $m$ many $j \in \{1,\ldots,m+1\}$ for which there exists at least one $i$ such that $x_{i,j-1}^* > x_{i,j}^*$. Then there exist $j, j' \in \{1,\ldots,m+1\}$ such that for every $i \in \{1,\ldots,m\}$, $x_{i,j-1}^* = x_{i,j}^*$ and $x_{i,j'-1}^* = x_{x,j'}^*$. Assume without loss of generality that $j' < j$ and therefore, $j > 1$. If $x_{i,j}^* = 0$ for every $i$, then by similar arguments as the previous case, $x_{i,m+1}^* = 0$ and the clauses $\neg x_{i,m}$ are satisfied. So suppose for the sake of contradiction that there exists a non-empty set $H_j = \{i : x_{i,j-1}^* = 1\} \neq \emptyset$. Note that $H_{j+1} := \{i : x_{i,j}^* = 1\} = H_j$, since $x_{i,j-1}^* = x_{i,j}^*$ for every $i$. Moreover, $\mathcal{I} \cap H_j = \emptyset$ since $x_{i,0}^* = 0$ for every $i \in \mathcal{I}$ and $\{x_{i,k}^*\}_k$ is non-increasing. Assume without loss of generality that $j$ is odd. The proof is identical in case $j$ is even. By definition, the set of vertices $H_{j+1} = H_j$ have a directed path to $S$ in $\mathcal{G}[H_j]$. Moreover, the set of vertices $H_j$ are those vertices in $H_{j-1}$

that have a bidirected path to $S$ in $\mathcal{G}[H_{j-1}]$ (here we used $j > 1$ for $H_{j-1}$ to be well-defined.) That is, $H_j$ is the connected component of $S$ in $\mathcal{G}[H_{j-1}]$. The latter implies that every vertex in $H_j$ has a bidirected path to $S$ in $\mathcal{G}[H_j]$. We proved that $H_j$ is a hedge formed for $S$, and $H_j \cap \mathcal{I} = \emptyset$. This contradicts with $\mathcal{I}$ intersecting with every hedge formed for $S$.

*Proof of 'only if:'* Suppose $\{x_{i,j}^*\}$ is a satisfying solution, where $x_{i,0}^* = 1$ for every $i \notin \mathcal{I}$. To prove $\mathcal{I}$ intersects every hedge formed for $S$, it suffices to show that there is no hedge formed for $S$ in $\mathcal{G}[V \setminus \mathcal{I}]$. Assume, for the sake of contradiction, that this is not the case. That is, there exists a hedge $H \subseteq V \setminus \mathcal{I}$ formed for $S$ in $\mathcal{G}$. Suppose for some $j \in \{1, \ldots, m+1\}$, it holds that $x_{i,j-1}^* = 1$ for every $v_i \in H$. We show that $x_{i,j}^* = 1$ for every $v_i \in H$. We consider the following two cases separately:

- Even $j$: for arbitrary $v_i, v_\ell \in H$ such that $\{v_i, v_\ell\} \in \overset{\leftrightarrow}{E}$, consider the clauses $(\neg x_{i,j-1} \vee x_{i,j} \vee \neg x_{\ell,j})$ and $(\neg x_{\ell,j-1} \vee x_{\ell,j} \vee \neg x_{i,j})$ that are in $F$ by construction for even $j$. Since $x_{i,j-1}^* = x_{\ell,j-1}^* = 1$, the expression $(x_{i,j}^* \vee \neg x_{\ell,j}^*) \wedge (x_{\ell,j}^* \vee \neg x_{i,j}^*)$ is satisfied; i.e., it evaluates to 'true.' The latter expression is equivalent to $(x_{i,j}^* \wedge x_{\ell,j}^*) \vee (\neg x_{i,j}^* \wedge \neg x_{\ell,j}^*)$, which implies that $x_{i,j}^* = x_{\ell,j}^*$. Note that $i, \ell$ were chosen arbitrarily. This implies that $x_{i,j}^*$ is equal for every $i$, since $H$ is a connected component through bidirected edges by definition of a hedge. Finally, since by construction, $x_{i,j} = 1$ for every $v_i \in S \subseteq H$, that equal value is 1. Therefore, $x_{i,j}^* = 1$ for every $i$ such that $v_i \in H$.

- Odd $j$: the proof is analogous to the case where $j$ is even. For arbitrary $v_i, v_\ell \in H$ such that $(v_i, v_\ell) \in \overset{\rightarrow}{E}$, consider the clause $(\neg x_{i,j-1} \vee x_{i,j} \vee \neg x_{\ell,j})$. Since $x_{i,j-1}^* = 1$, the expression $(x_{i,j}^* \vee \neg x_{\ell,j}^*)$ is satisfied; i.e., it evaluates to 'true.' The latter implies that if $x_{\ell,j}^* = 1$ and $v_i$ has a directed edge to $v_j$, then $x_{i,j}^* = 1$. Using the same argument recursively, if $x_{\ell,j}^* = 1$ and $v_i$ has a directed 'path' to $v_j$ in $\mathcal{G}[H]$, then $x_{i,j}^* = 1$. By construction, $x_{\ell,j}^* = 1$ for every $v_\ell \in S$, and by definition of a hedge, every vertex $v_i \in H$ has a directed path to $S$. As a result, $x_{i,j}^* = 1$ for every $v_i \in H$.

Since for every $v_i \in H$, $x_{i,0}^* = 1$, using the arguments above, by induction, $x_{i,m+1}^* = 1$ for every $v_i \in H$. However, this contradicts the fact that $\{x_{i,j}^*\}$ satisfies $F$, since $F$ includes the clauses $\neg x_{i,m+1}$ for every $i$. $\square$

**Lemma 1.** *Let $\mathcal{S} = \{S_1, \ldots, S_r\}$ be the set of districts of $S$, where $S = \mathrm{Anc}_{V \setminus X}(Y)$. There exists an intervention set family $\mathcal{I}^*$ of size $|\mathcal{S}| = r$ that is optimal for identifying $\mathbb{P}_X(Y)$.*

*Proof.* Let $\mathcal{I}$ be an optimizer of Eq. (1). By Proposition 1, for every $S_i \in \mathcal{S}$, there exists $\mathcal{I}_j \in \mathcal{I}$ that hits every hedge formed for $S_i$. The set of such $\mathcal{I}_j$s is a subset of at most size $r$ of $\mathcal{I}$, which implies that $|\mathcal{I}| \leq r$ since $\mathcal{I}$ is optimal. If $|\mathcal{I}| = r$, the claim is trivial. If $|\mathcal{I}| < r$, then simply add $(r - |\mathcal{I}|)$ empty intervention sets to $\mathcal{I}$ to form an intervention set family $\mathcal{I}^*$ with the same cost which is an optimal solution to Eq. (1). $\square$

**Theorem 2.** *Suppose $\mathcal{G}$, a set of its vertices $S$ with districts $\mathcal{S} = \{S_1, \ldots, S_r\}$, and an intervention set family 2 $\mathcal{I} = \{\mathcal{I}_1, \ldots, \mathcal{I}_r\}$ are given. Define $m_\ell = |\mathcal{H}_\mathcal{G}(S_\ell) \setminus S_\ell|$, i.e., the cardinality of the hedge hull of $S_\ell$ excluding $S_\ell$ itself. The SAT formula $F$ constructed by Algorithm 3 has a satisfying solution $\{x_{i,0,k}^*\} \cup \{x_{i,j,k,\ell}^*\} \cup \{z_{k,\ell}^*\}$ where for every $\ell \in \{1, \ldots, r\}$, there exists $k \in \{1, \ldots, r\}$ such that (i) $z_{k,\ell}^* = 1$, (ii) $x_{i,0,k}^* = 0$ for every $i \in \mathcal{I}_k$, and (iii) $x_{i,0,k}^* = 1$ for every $i \in \{1, \ldots, m_\ell\} \setminus \mathcal{I}_k$, if and only if $\mathcal{I}$ is a feasible solution to optimization of Eq. (1).*

*Proof.* The proof is identical to that of Theorem 1 with necessary adaptations.

*Proof of 'if:'* Suppose $\mathcal{I}$ is a solution to Eq. (1). From Proposition 1, for every $S_\ell \in \mathcal{S}$, there exists $k$ such that $\mathcal{I}_k$ hits every hedge formed for $S_\ell$. Assign $z_{k,\ell}^* = 1$ and $z_{k',\ell}^* = 0$ for every other $k' \neq k$, thereby satisfying every clause that includes $\neg z_{k',\ell}$, $k' \neq k$. So it suffices to assign values to other variables so that clauses including $\neg z_{k,\ell}$. Since $z_{k,\ell} = 1$, these clauses reduce to $(\neg x_{i,j-1,k,\ell} \vee x_{i,j,k,\ell} \vee \neg x_{p,j,k,\ell})$ (see lines 16, 19, or 20.) These clauses are exactly in the form of 3-SAT clauses as in the single-district case procedure. An assignment exactly parallel to the proof of Theorem 1 satisfies these clauses.

*Proof of 'only if:'* Suppose $\{x_{i,0,k}^*\} \cup \{x_{i,j,k,\ell}^*\}$ is a satisfying solution, where for some $k, \ell \in \{1, \ldots, r\}$, it holds that $z_{k,\ell}^* = 1$ and $x_{i,0,k}^* = 1$ for every $i \notin \mathcal{I}_k$. We show that $\mathcal{I}_k$ hits every hedge formed for $S_\ell$. Since such a $k$ exists for every $S_\ell$, we will conclude that $\mathcal{I}$ is feasible for Eq. (1) by Proposition 1. Finally, to show that $\mathcal{I}_k$ hits every hedge formed for $S_\ell$, note that satisfiability of all clauses containing the literal $\neg z_{k,\ell}^*$ reduces to the satisfiability of $(\neg x_{i,j-1,k,\ell} \lor x_{i,j,k,\ell} \lor \neg x_{p,j,k,\ell})$ (see lines 16, 19, or 20), and the same arguments as in the proof of Theorem 1 apply. $\qquad\square$

## C.2 Results of Section 4

**Lemma 2.** *Let $X, Y$ be two disjoint sets of vertices in $\mathcal{G}$ such that $X$ is minimal as defined above. Set $Z \subseteq V$ is a generalized adjustment set for $\mathbb{P}_X(Y)$ under intervention $\mathcal{I}$ if (i) $Z \subseteq \mathrm{Anc}(S)$, and (ii) $Z$ is a vertex cut[9] between $S$ and $\mathrm{Pa}(S)$ in $(\mathcal{G}_{\overline{\mathcal{I}}\mathrm{Pa}(S)})^m$, where $S = \mathrm{Anc}_{V \setminus X}(Y)$, and $\mathcal{G}_{\overline{\mathcal{I}}\mathrm{Pa}(S)}$ is the ADMG resulting from omitting all edges incoming to $\mathcal{I}$ and all edges outgoing of $\mathrm{Pa}(S)$.*

*Proof.* Define $S = \mathrm{Anc}_{V \setminus X}(Y)$. First, we show that $\mathrm{Pa}(S) \subseteq X$. Assume the contrary, i.e., there is a vertex $w \in \mathrm{Pa}(S) \setminus X$. Clearly $w$ has a directed path to $S$ (a direct edge) that does not go through $X$. This implies that $w \in \mathrm{Anc}_{V \setminus X}(S)$, and since by definition, $S = \mathrm{Anc}_{V \setminus X}(Y)$, $w \in \mathrm{Anc}_{V \setminus X}(Y) = S$. However, the latter contradicts with $w \in \mathrm{Pa}(S)$. Second, we note that from the third rule of do calculus [Pearl, 2009], $\mathbb{P}_W(S) = \mathbb{P}_{\mathrm{Pa}(S)}(S)$ for any $W \supseteq \mathrm{Pa}(S)$. Combining the two arguments, we have the following:

$$\mathbb{P}_W(S) = \mathbb{P}_X(S), \quad \forall W \supseteq \mathrm{Pa}(S). \tag{6}$$

To proceed, we will use the following proposition.

**Proposition 3** (Lauritzen et al., 1990). *Let $S, R,$ and $Z$ be disjoint subsets of vertices in a directed acyclic graph $\mathcal{G}$. Then $Z$ d-separates $S$ from $R$ if and only if $Z$ is a vertex cut between $S$ and $R$ in $(\mathcal{G}[\mathrm{Anc}(S \cup R \cup Z)])^m$.*

Choose $R = \mathrm{Pa}(S)$ in the proposition above. Since $Z \subseteq \mathrm{Anc}(S)$, we have that $\mathrm{Anc}(S \cup R \cup Z) = \mathrm{Anc}(S)$. From condition (ii) in the lemma, $Z$ is a vertex cut between $S$ and $R$ in $(\mathcal{G}_{\overline{\mathcal{I}}\mathrm{Pa}(S)})^m$, which implies it is also a vertex cut in $(\mathcal{G}_{\overline{\mathcal{I}}\mathrm{Pa}(S)}[\mathrm{Anc}(S)])^m$, as every path in the latter graph exists in $(\mathcal{G}_{\overline{\mathcal{I}}\mathrm{Pa}(S)})^m$. Using the proposition above, $Z$ d-separates $S$ and $\mathrm{Pa}(S)$ in $\mathcal{G}_{\overline{\mathcal{I}}\mathrm{Pa}(S)}$. This is to say, $Z$ blocks all non-causal paths from $\mathrm{Pa}(S)$ to $S$ in $\mathcal{G}_{\overline{\mathcal{I}}}$, and it clearly has no elements that are descendants of $\mathrm{Pa}(S)$. Therefore, $Z$ satisfies the adjustment criterion of Shpitser et al. [2010] w.r.t. $\mathbb{P}_{\mathrm{Pa}(S)}(S)$ in $\mathcal{G}_{\overline{\mathcal{I}}}$. That is, the following holds:

$$\mathbb{P}_{\mathcal{I} \cup \mathrm{Pa}(S)}(S) = \mathbb{E}_{\mathbb{P}_{\mathcal{I}}}[\mathbb{P}_{\mathcal{I}}(S \mid \mathrm{Pa}(S), Z)],$$

where the expectation is w.r.t. $\mathbb{P}_{\mathcal{I}}(Z)$. Choosing $W = \mathcal{I} \cup \mathrm{Pa}(S)$ in Eq. (6), we get

$$\mathbb{P}_X(S) = \mathbb{E}_{\mathbb{P}_{\mathcal{I}}}[\mathbb{P}_{\mathcal{I}}(S \mid \mathrm{Pa}(S), Z)].$$

Marginalizing $S \setminus Y$ out in both sides of the equation above, we have

$$\mathbb{P}_X(Y) = \mathbb{E}_{\mathbb{P}_{\mathcal{I}}}[\mathbb{P}_{\mathcal{I}}(Y \mid \mathrm{Pa}(S), Z)].$$

The last step of the proof is to show that $\mathrm{Pa}(S) = X$. We already showed that $\mathrm{Pa}(S) \subseteq X$. For the other direction, we will use the minimality of $X$. Suppose to the contrary that $x \in X \setminus \mathrm{Pa}(S)$. We first show that every causal path from $x$ to $Y$ goes through $X \setminus \{x\}$. Suppose not. Then take a causal path $x, s_1, \ldots, s_m, y$ be a causal path from $x$ to $y \in Y$. Note that $s_1$ has a causal path to $Y$ that does not go through $X$. By definition, $s_1 \in S$, which implies $x \in \mathrm{Pa}(S)$, which is a contradiction. Therefore, every causal path from $x$ to $Y$ goes through $X \setminus \{x\}$, and consequently, there is no causal path from $x$ to $Y$ in $\mathcal{G}_{\overline{X}}$. Clearly there is no *backdoor* path either. Every other path has a collider on it, and therefore is blocked by $X \setminus \{x\}$ – note that none of these can be colliders in $\mathcal{G}_{\overline{X}}$. Therefore, $\{x\}$ is d-separated from $Y$ given $X \setminus \{x\}$ in $\mathcal{G}_{\overline{X}}$, which contradicts the minimality of $X$ w.r.t. the third rule of do calculus. This shows $X \subseteq \mathrm{Pa}(S)$, completing the proof. $\qquad\square$

---

[9]This corresponds to $Z$ blocking all the backdoor paths between $\mathrm{Pa}(S)$ and $S$, in the modified graph $\mathcal{G}_{\overline{\mathcal{I}}}$.

**Theorem 3.** *Let $(\mathcal{I}, Z)$ be the output returned by Algorithm 1 for the query $\mathbb{P}_X(Y)$. Then,*
*- $Z$ is a generalized adjustment set for $\mathbb{P}_X(Y)$ under intervention $\mathcal{I}$.*
*- $\mathcal{I}$ is the minimum-cost intervention for which there exists a generalized adjustment set based on the graphical criterion of Lemma 2.*

*Proof.* For the first part, using Lemma 2, it suffices to show that $Z$ is a vertex cut between $S$ and $\mathrm{Pa}(S)$ in $(\mathcal{G}_{\overline{\mathcal{I}}\mathrm{Pa}(S)})^m$. Suppose not. That is, there exists a path from $S$ to $\mathrm{Pa}(S)$ in $(\mathcal{G}_{\overline{\mathcal{I}}\mathrm{Pa}(S)})^m$ that does not pass through any member of $Z$. Let $P = s, v^1, \ldots, v^l, x$ represent this path, where $s \in S$ and $x \in \mathrm{Pa}(S)$. We construct a corresponding path $P'$ in $\mathcal{G}^{vc}$ as follows. The first vertex on $P'$ is $s_1$, which corresponds to the first vertex on $P$, $s$. We then walk along $P$ and add a path to $P'$ corresponding to each edge we traverse on $P$ as follows. Consider this edge to be $\{v, w\}$ – for instance, the first edge would be $\{s, v^1\}$. By definition of $(\mathcal{G}_{\overline{\mathcal{I}}\mathrm{Pa}(S)})^m$, for every pair of adjacent vertices $v, w$ on the path $P$, one of the following holds: (i) $v \to w$ in $(\mathcal{G}_{\overline{\mathcal{I}}\mathrm{Pa}(S)})^d$, (ii) $v \leftarrow w$ in $(\mathcal{G}_{\overline{\mathcal{I}}\mathrm{Pa}(S)})^d$, or (iii) $v$ and $w$ have a common child $t$ in $(\mathcal{G}_{\overline{\mathcal{I}}\mathrm{Pa}(S)})^d$. In case (i), we add $v_1, w_2, w_1$. In case (ii), we add $v_2, w_1$. Finally, in case (iii), we add $v_1, t_2, w_1$ to $P'$. We continue this procedure until we traverse all edges on $P$. The last vertex on $P'$ is $x_2$, as x has no children in $(\mathcal{G}_{\overline{\mathcal{I}}\mathrm{Pa}(S)})^d$. Finally we add $x_1$ to this path, as $x_2$ and $x_1$ are always connected by construction. Note that by construction of $P'$, any vertex that appears with index 2 has a parent in $v \to w$ in $(\mathcal{G}_{\overline{\mathcal{I}}\mathrm{Pa}(S)})^d$, and therefore is not a member of $\mathcal{I}$. Hence, $\{v_2 : v \in \mathcal{I}\}$ does not intersect with $P'$. Further, $\{v_1 : v \in Z\}$ does not intersect with $P'$ either, as none of the vertices appearing on $P'$ correspond to $Z$. This is to say that $J = \{v_2 : v \in \mathcal{I}\} \cup \{v_1 : v \in Z\}$, which is the solution obtained by Algorithm 1 in line 10, does not cut the path $P'$. This contradicts with $J$ being a vertex cut.

For the second part, let $(\mathcal{I}', Z')$ be so that $Z'$ is a vertex cut between $S$ and $\mathrm{Pa}(S)$ in $(\mathcal{G}_{\overline{\mathcal{I}'}\mathrm{Pa}(S)})^m$, and $\mathcal{I}'$ induces a lower cost than $\mathcal{I}$; that is, $C(\mathcal{I}') < C(\mathcal{I})$. Define $J' = \{v_2 : v \in \mathcal{I}'\} \cup \{v_1 : v \in Z'\}$. Clearly, the cost of $J'$ is equal to $C(\mathcal{I}')$, which is lower than the cost of minimum vertex cut found in line 10 of Algorithm 1. It suffices to show that $J'$ is also a vertex cut between $\{x_1 : x \in \mathrm{Pa}(S)\}$ and $\{s_1 : s \in S\}$ in $\mathcal{G}^{vc}$ to arrive at a contradiction. Suppose not; that is, there is a path $P = s_1, \ldots, x_1$ on $\mathcal{G}^{vc}$ that $J' = \{v_2 : v \in \mathcal{I}'\} \cup \{v_1 : v \in Z'\}$ does not intersect. None of the vertices with index 2 on $P$ belong to $\mathcal{I}'$, and none of the vertices with index 1 belong to $Z'$. Analogous to the first part, we construct a corresponding path $P'$ – this time in $(\mathcal{G}_{\overline{\mathcal{I}'}\mathrm{Pa}(S)})^m$. The starting vertex on $P'$ is $s$, which corresponds to $s_1$, the initial vertex on $P$. Let us imagine a cursor on $s_1$. We then sequentially build $P'$ by traversing $P$ as follows. We always look at sequences starting with $v_1$ (where the cursor is located): when the sequence is of the form $v_1, w_2, w_1$ or $v_1, v_2, w_1$, we add $w$ to $P'$, and move the cursor to $w_1$; however, when the sequence is of the form $v_1, w_2, r_1$, we add $r_1$ to $P'$ and move the cursor to $r_1$. By construction of $\mathcal{G}^{vc}$, no other sequence is possible – note that there are no edges between $v_1$ and $w_1$ or $v_2$ and $w_2$ where $v$ and $w$ are distinct. Since none of the vertices with index 2 on $P$ belong to $\mathcal{I}$, in the first case, the corresponding edge $v \leftarrow w$ or $v \to w$ is present in $(\mathcal{G}_{\overline{\mathcal{I}'}\mathrm{Pa}(S)})^d$ and consequently, the edge $\{v, w\}$ is present in $(\mathcal{G}_{\overline{\mathcal{I}'}\mathrm{Pa}(S)})^m$; and in the latter case, both edges $v \to w$ and $w \leftarrow r$ are present in $(\mathcal{G}_{\overline{\mathcal{I}'}\mathrm{Pa}(S)})^d$ and consequently, the edge $\{v, r\}$ is present in $(\mathcal{G}_{\overline{\mathcal{I}'}\mathrm{Pa}(S)})^m$. $P'$ is therefore a path between $S$ and $\mathrm{Pa}(S)$ in $(\mathcal{G}_{\overline{\mathcal{I}'}\mathrm{Pa}(S)})^m$. Notice that by construction, only those vertices appear on $P'$ that their corresponding vertex with index 1 appears on $P$ – the *cursor* always stays on vertices with index 1. As argued above, none of such vertices belong to $Z'$, which means $Z'$ does not intersect with $P'$ which is a path from $S$ to $\mathrm{Pa}(S)$ in $(\mathcal{G}_{\overline{\mathcal{I}'}\mathrm{Pa}(S)})^m$. This contradicts with $Z'$ being a vertex cut. $\square$

### C.2.1 Proof of Remark 3

*Proof.* Since the algorithms are run in the hedge hull of $S$, assume without loss of generality that $V = \mathcal{H}_\mathcal{G}(S)$, i.e., $V$ is the hedge hull of $S$. From Theorem 3, Algorithm 1 finds the optimal (minimum-cost) intervention $\mathcal{I}$ such that there exists a set $Z \subseteq V$ that is a vertex cut between $S$ and $\mathrm{Pa}(S)$ in $(\mathcal{G}_{\overline{\mathcal{I}}\mathrm{Pa}(S)})^m$. To prove that the cost of the solution returned by Algorithm 1 is always smaller than or equal to that of heuristic algorithm 1 of Akbari et al. [2022], it suffices to show that the solution of their algorithm is a feasible point for the statement above. That is, denoting the output

of heuristic algorithm 1 of Akbari et al. [2022] by $I_1$, we will show that there exist sets $Z_1 \subseteq V$ such that it is a vertex cut between $S$ and $\mathrm{Pa}(S)$ in $(\mathcal{G}_{\overline{I_1}\mathrm{Pa}(S)})^m$.

Heuristic algorithm 1: This algorithm returns an intervention set $I_1$ such that there is no bidirected path from $\mathrm{Pa}(S)$ to $S$ in $\mathcal{G}_{\overline{I_1}}$. We claim $Z_1 = V \setminus \mathrm{Pa}(S) \setminus S$ satisfies the criterion above. To prove this, consider an arbitrary path $P$ between $S$ and $\mathrm{Pa}(S)$ in $(\mathcal{G}_{\overline{I_1}\mathrm{Pa}(S)})^m$. If there is an observed vertex $v \in V$ on $P$, this vertex is included in $Z_1$ and separates the path. So it suffices to show that there is no path $P$ between $S$ and $\mathrm{Pa}(S)$ in $(\mathcal{G}_{\overline{I_1}\mathrm{Pa}(S)})^m$ where all the intermediate vertices on $P$ correspond to unobserved confounders. Suppose there is. That is, $P = x, u_1, \ldots, u_m, s$, where $x \in \mathrm{Pa}(S)$ and $\{u_i\}_{i=1}^m$ are unobserved. Since $u_i$ is not connected to $u_j$ in $(\mathcal{G}_{\overline{I_1}\mathrm{Pa}(S)})^d$, it must be the case that any $u_i$ and $u_{i+1}$ have a common child $v_{i,j}$ in $(\mathcal{G}_{\overline{I_1}\mathrm{Pa}(S)})^d$. This is to say, there is a path $x \leftarrow u_1 \rightarrow v_{1,2} \leftarrow u_2 \rightarrow \cdots \rightarrow v_{m-1,m} \leftarrow u_m \rightarrow s$ in $(\mathcal{G}_{\overline{I_1}\mathrm{Pa}(S)})^d$, which corresponds to the bidirected path $x, v_{1,2}, \ldots, v_{m-1,m}, s$ in $\mathcal{G}_{\overline{I_1}}$. This contradicts with the fact that there is no bidirected path between $\mathrm{Pa}(S)$ and $S$ in $\mathcal{G}_{\overline{I_1}}$. $\qquad\square$

### C.3 Results in the appendices

**Lemma 3.** *For any district $S \subseteq V$, the function $f_S : 2^{V \setminus S} \to \mathbb{Z}^{\leq 0}$ is submodular.*

*Proof.* Take two distinct vertices $\{x, y\} \in V \setminus S$ and an arbitrary set $I \subset V \setminus S \setminus \{x, y\}$. It suffices to show that

$$f_S(I \cup \{x, y\}) - f_S(I \cup \{y\}) \leq f_S(I \cup \{x\}) - f_S(I).$$

By definition, the right hand side is the number of hedges $H \subseteq V \setminus I$ formed for $S$ such that $x \notin H$. Similarly, the left hand side counts the number of hedges $H \subseteq V \setminus (I \cup \{y\})$ formed for $S$ such that $x \notin H$. The inequality holds because the set of hedges counted by the left hand side is a subset of that on the right hand side. $\qquad\square$

**Proposition 4.** *The combinatorial optimization of Eq. (2) is equivalent to the following unconstrained submodular optimization problem.*

$$\mathcal{I}^* = \operatorname*{argmax}_{\mathcal{I} \subseteq V \setminus S} \left( f_S(\mathcal{I}) - \frac{\sum_{v \in \mathcal{I}} C(v)}{1 + \sum_{v \in V \setminus S} C(v)} \right). \tag{7}$$

*Proof.* The submodularity of the objective function follows from Lemma 3. To show the equivalence of the two optimization problems, we show that a maximizer $\mathcal{I}^*$ of Eq. (7) (i) hits every hedge formed for $S$, and (ii) has the optimal cost among such sets.

*Proof of '(i):'* Note that $f_S(\mathcal{I}) = 0$ if and only if there are no hedges formed for $S$ in $\mathcal{G}[V \setminus \mathcal{I}^*]$, or equivalently, $\mathcal{I}$ hits every hedge formed for $S$. So it suffices to show that $f_S(\mathcal{I}^*) = 0$ for every maximizer $\mathcal{I}^*$ of Eq. (7). To this end, first note that

$$f_S(V \setminus S) - \frac{\sum_{v \in V \setminus S} C(v)}{1 + \sum_{v \in V \setminus S} C(v)} = 0 - 1 + \frac{1}{1 + \sum_{v \in V \setminus S} C(v)} > -1,$$

which implies that

$$f_S(\mathcal{I}^*) - \frac{\sum_{v \in \mathcal{I}^*} C(v)}{1 + \sum_{v \in V \setminus S} C(v)} > -1.$$

On the other hand, clearly $\frac{\sum_{v \in \mathcal{I}^*} C(v)}{1 + \sum_{v \in V \setminus S} C(v)} \geq 0$, which combined with the inequality above implies $f_S(\mathcal{I}^*) > -1$. Since $f_S(\mathcal{I}^*) \in \mathbb{Z}^{\leq 0}$, it is only possible that $f_S(\mathcal{I}^*) = 0$.

*Proof of '(ii):'* We showed that $f_S(\mathcal{I}^*) = 0$. So $\mathcal{I}^*$ maximizes $\frac{\sum_{v \in \mathcal{I}} C(v)}{1 + \sum_{v \in V \setminus S} C(v)}$ among all those $I$ such that $f_S(I) = 0$. Since the denominator is a constant, this is equivalent to minimizing $C(I) = \sum_{v \in I} C(v)$ among all those $I$ that hit all the hedges formed for $S$, which matches the optimization of Eq. (2). $\qquad\square$

# D  Alternative Formulations

## D.1  Min-cost intervention as a submodular function maximization problem

In this Section, we reformulate the minimum-cost intervention design as a submodular optimization problem. Submodular functions exhibit a property akin to diminishing returns: the incremental gain from adding an element to a set decreases as the set grows [Nemhauser et al., 1978].

**Definition 5.** *A function $f : 2^V \to \mathbb{R}$ is submodular if for all $A \subseteq B \subseteq V$ and $v \in V \setminus B$, we have that $f(A \cup \{v\}) - f(A) \geq f(B \cup \{v\}) - f(B)$.*

Given an ADMG $\mathcal{G} = \langle V, \overrightarrow{E}, \overleftrightarrow{E} \rangle$ a district $S$ in $\mathcal{G}$, and an arbitrary set $\mathcal{I} \subseteq V \setminus S$, we define $f_S(\mathcal{I})$ as the negative count of hedges formed for $S$ in $\mathcal{G}[V \setminus \mathcal{I}]$.

**Lemma 3.** *For any district $S \subseteq V$, the function $f_S : 2^{V \setminus S} \to \mathbb{Z}^{\leq 0}$ is submodular.*

Note that $g_S : 2^{V \setminus S} \to \mathbb{R}$, where $g_S(\mathcal{I}) \coloneqq f_S(\mathcal{I}) + \alpha \sum_{v \in \mathcal{I}} C(v)$, and $\alpha$ is an arbitrary constant, is also submodular as the second component is a modular function (similar definition as in 5 only with equality instead of inequality.).

**Proposition 4.** *The combinatorial optimization of Eq.* (2) *is equivalent to the following unconstrained submodular optimization problem.*

$$\mathcal{I}^* = \operatorname*{argmax}_{\mathcal{I} \subseteq V \setminus S} \left( f_S(\mathcal{I}) - \frac{\sum_{v \in \mathcal{I}} C(v)}{1 + \sum_{v \in V \setminus S} C(v)} \right). \tag{7}$$

## D.2  Min-cost intervention as an RL problem

We model the MCID problem given a graph $\mathcal{G} = (V, E)$ and $S \subset V$, as a Markov decision process (MDP), where a vertex is removed in each step $t$ until there are no hedges left. The goal is to minimize the cost of the removed vertices (i.e., intervention set). Naturally, the action space is the set of vertices, $V$ and the state space is the set of all subsets of $V$. More precisely, let $s_t$ and $a_t$ denote the state and the action of the MDP at iteration $t$, respectively. Then, $s_t$ is the hedge hull for $S$ from the remaining vertices at time $t$, and action $a_t$ is the vertex that will be removed from $V_t$ in that iteration. Consequently, the state transition due to action $a_t$ is $s_{t+1} = \mathcal{H}_{\text{hull}}(V_t \setminus \{a_t\})$. The immediate reward of selecting action $a_t$ at state $s_t$ will be the negative of the cost of removing (i.e., intervening on) $a_t$, given by

$$r(s_t, a_t) = -C(a_t).$$

The MDP terminates when there are no hedges left and the hedge hull of the remaining vertices is empty (i.e., $s_t = \emptyset$). The goal is to find a policy $\pi$ that maximizes sum of the rewards until the termination of the MDP. Formally, the goal is to solve

$$\operatorname*{argmax}_{\pi} \left[ \sum_{t=1}^{T} r(s_t, a_t) \right],$$

where $s_1 = V$ and $T$ is the time step at which the MDP terminates (i.e., $s_T = \emptyset$).

