# OpenReview forum: "Fast Proxy Experiment Design for Causal Effect Identification"
_NeurIPS.cc/2024/Conference — NeurIPS 2024 poster_

### Official Review · Reviewer_tZFY · 2024-07-08

**Soundness:** 1
**Presentation:** 1
**Contribution:** 2
**Rating:** 4
**Confidence:** 5

**Summary:**

This paper provide a computationally efficient algorithm for finding the sets of variables $\mathbf{Z}_1,\cdots,\mathbf{Z}_m$ that achieve the minimum intervention cost, allowing for $P(\mathbf{y} \mid \operatorname{do}(\mathbf{x}))$ to be identifiable from $\{P(\mathbf{V} \mid \operatorname{do}(\mathbf{Z}_i)): i=1,\cdots,m\}$.

**Strengths:**

1. The research problem is interesting and practical.
2. Experiments conducted on the large graph provide empirical evidence supporting the proposed method.

**Weaknesses:**

__1. Misleading and wrong statements__

This paper contains some statements that could mislead readers. First, there is a sentence that doesn’t acknowledge the previous works properly.

> A middle ground between the two extremes of observational and experimental approaches was introduced by Akbari et al. [2022]
>

The middle ground, which uses the fusion of observational and experimental data to identify the causal effect, is introduced by [Bareinboim and Pearl (2012)](https://arxiv.org/pdf/1210.4842), not Akbari et al., 2022. This sentence could mislead readers by ignoring existing works on the middle ground.

Second, this paper incorrectly defines some existing notions in theories of causal effect identification or gives new names to already existing concepts.

- For example, a “district” is a *maximal* set of vertices connected by bidirected edges [[Richardson et al., 2023](https://arxiv.org/abs/1701.06686) or [Shpitser et al., 2014](https://web.archive.org/web/20181102234555id_/https://www.jstage.jst.go.jp/article/bhmk/41/1/41_3/_pdf)]. Therefore, given a (sub-)graph $\mathcal{G}(T)$ and a set of vertices $S$, there is only one district containing $S$ in $\mathcal{G}(T)$. However, this paper wrongly defines “district” without invoking *maximality.*
- Another example is “hedge hull.” Acknowledging the original definition of the district, the hedge hull is just a hedge defined by [Shpitser et al., 2008](https://jmlr.org/papers/volume9/shpitser08a/shpitser08a.pdf) and [Shpitser al el., 2012](https://arxiv.org/pdf/1202.3763), which is an ancestor of $S$ and a district at the same time. I don’t see why the authors redefine the original “hedge” as “hedge hull.” It only raises confusion in comprehending the paper.

Third, some of statements are not sound. Specifically, the following sentence in line 133-134 seems to be incorrect.

> Note that there is no hedge formed for $S_{\ell}$ in $\mathcal{G}[V \setminus \mathcal{I}_k]$ if and only if \mathcal{I}_k hits every hedge of $S_{\ell}$ (i.e.,for any hedge $W \in H_{\mathcal{G}}(S_{\ell})$,$\mathcal{I}_k \cap W \ neq \emptyset$).
>

For example,

* Consider the case where $S_{\ell}$ is identifiable in $\mathcal{G}$. Then, there must be no hedge formed for $S_{\ell}$ in $\mathcal{G}$. However, the condition where $\mathcal{I}_k$ and $W$ is non-overlapped is violated.

* Another case is when $S_{\ell}$ itself is a hedge. Then, with $\mathcal{I}_k = \emptyset$, the sentence doesn't hold.

Also, The following sentence is false:

> The query of interest is identifiable if and only if HG (S) = S, that is, the hedge hull of S contains no other vertices.
>

Consider the graph $\mathcal{G} = ( W → R → X → Z → Y, W ↔ X, W↔Z, X↔Y )$. Here, $S = \operatorname{Anc}_{V \setminus X}(Y) = (Z,Y)$, but the hedge hull found from Algorithm 2 is $(X,Z,Y)$. However, the causal effect P(Y | do(X)) is identifiable from $\mathcal{G}$.

---

__2. Weak presentation__

First, the paper weakly exemplifies the SAT Construction Procedure. Without providing detailed examples of how 3-SAT can be constructed using the graphs in Figure 1, it's hard to understand how the problem is reformulated using 3-SAT.

Second, many important results are omitted (e.g., a detailed example for 3-SAT construction and IP formulation) and presented in the supplementary document (e.g., Algorithms 2 and 3). Specifically, it's impossible to comprehend this paper without reading Section B. Given that reading the supplementary document is not mandatory for reviewers and readers, I encourage the authors to reorganize the paper to make it more comprehensible.


---

__3. Lack of analysis \& misleading contribution__

To my knowledge, the exact solution for integer programming is NP-hard, and only heuristic methods are available for efficiently solving the problem. A natural question is the gap between the exact solution and these approximated solutions.

Also, the authors need to be more cautious in presenting their contribution. The fast experiment design was achieved by heuristically solving the problem, and the exact solution remains NP-hard. However, the paper is presented as if the NP-hard problem has been solved just by reformulating the problem.

---

__4.Minor comments__

1. The following sentence is not parsable. Is “win” a typo?

> Moreover, Akbari et al. [2022]’s algorithm was tailored to a specific class of causal effects win which the effect of interest is a functional of an interventional distribution where the intervention is made on every variable except one district of the causal graph
>

**Questions:**

1. In line 109, is the condition S \subsetneq W is necessary?

**Limitations:**

1. This paper still solves the heuristic solution, instead of the exact solution. The exact solution of the problem still remains NP-hard.

---

> ### Author Rebuttal · Authors · 2024-08-06
>
> We sincerely thank the reviewer for their thorough reading of our paper and their detailed feedback. We have addressed each comment and question below.
> ## Weaknesses:
> ### W1:
> We acknowledge this comment. We have made our statements more precise to clarify that data fusion for causal identification goes way back (as we ourselves build upon a follow-up to Bareinboim and Pearl, 2012, namely Lee et al., 2020), and we are revising our manuscript to include a comprehensive literature review on this; but that the idea of surrogate experiment design for causal identification was put forward by Akbari et al., 2022.
> ### W2:
> We indeed slightly diverged from the original definitions of hedge and district, particularly by not imposing maximality. This was done solely to simplify the presentation of our work and enhance readability. These adjusted definitions are what make our arguments sound. However, we acknowledge that this can be confusing for certain readers. To address this, we are updating our manuscript to use alternative names or be more explicit about the altered definitions wherever possible. We have additionally included proper citations to the relevant papers by Richardson et al. 2023 and Shpitser et al. 2008, 2012, 2014.
> ### W3:
> If $S_\ell$ is identifiable in $\mathcal{G}$, i.e., there is no hedge formed for$S_\ell$ in $\mathcal{G}$, then $H_\mathcal{G}(S_\ell) = \emptyset$. Thus, the statement *for any $W$ in $H_\mathcal{G}(S_\ell)$, $\mathcal{I}_k \cap W \neq \emptyset$* is true by the concept of vacuous truth, as the antecedent (i.e., $\forall W \in H_\mathcal{G}(S_\ell)$) can never be satisfied.
>
> As for your second example, we are not sure what you mean by "when $S_\ell$ is a hedge." By our definition of hedge (Def. 2), $S_\ell$ can never be its own hedge. If you meant that $S_\ell$ is within a hedge for another district $S_k$, then our statement is still valid. For instance, consider the graph of Fig. 2(d), where $S_2$ is involved in a hedge formed for $S_1$, and our statement is still valid with $I_k=\\{S_2\\}$.
> ### W4:
> Our Def. 3 defines the hedge hull for a (single) district. Right after this definition, we define the hedge hull of an arbitrary set $S$ as the union of the hedge hulls of each of its districts. In your specific example, $S=\\{Z,Y\\}$ consists of two districts, $S_1=\\{Z\\}$ and $S_2=\\{Y\\}$. Following Def. 3, the hedge hull of $S_1$ is $S_1$ itself, and the same goes for $S_2$. Therefore, $\mathcal{H}_{\mathcal{G}}(S)=S_1\cup S_2=S$, which indicates that $P(Y\mid do(X))$ is identifiable, as expected.
>
> Our paper first covers the case where $S$ comprises a single district, then generalizes the results to the case of arbitrary sets (see lines 157-158,). As such, Alg. 2 is designed for single-district $S$. We have made this clearer in the revised version. In particular, we have explicitly stated that if $S$ comprises more than one district, Alg. 2 is run on each district separately to compute the hedge hull of $S$.
> ### W5:
> Due to space constraints, we had to move the details of the SAT construction to Appendix B.2. However, with the extra page allocated for the camera-ready, we are relocating these construction details to the main text. We have also provided a walkthrough of how the SAT expression can be constructed using a toy graph from Figure 2. This will clarify the SAT reformulation. We will also add the DFS encoding of a graph as a SAT problem in the appendix and reference it in the main text, aiding readers, especially those new to SAT constructions.​
>
> We have condensed Algorithms 2 and 3, and brought them into the main text, with a brief explanation of the rationale behind them to aid the reader in understanding the work more easily.
> ### W6:
> It’s important to clarify that the MCID problem is NP-hard, as shown by Akbari et al. 2022, with the best achievable polynomial-time approximation being within a $\log(n)$ factor, where $n$ is the number of vertices. Thus, the worst-case time complexity of our reformulations is expected to be exponential in $n$.
>
> Our reformulations not only solve the problem exactly, but also provide a significant speedup of 1,000 to 100,000,000 times for the single-district case.
>
> Moreover, for the multiple-district case, our approach is only quadratic in the number of districts (theoretically guaranteed), unlike the previous method which required iterating over an exponential number of partition sets. This efficiency is demonstrated in Fig. 3b, showing speedups of 50 to 10,000 times for a modest number of districts.
>
> We believe therefore the term “fast” is justified. Although the exact solution remains exponential in the worst case, our reformulations make solving these problems practically feasible and significantly more efficient, especially for large graphs.
>
> Another significant contribution is our novel min-cost experiment design for generalized adjustment (Section 4), for which we provide a polynomial-time algorithm.
>
> These distinctions will be more clearly presented in the revised text to accurately reflect our contributions and their practical implications.
> ## Questions (Q1):
> Yes, the condition $S \subsetneq W$ ensures that $S$ cannot form a hedge for itself, in line with Shpitser and Pearl (2006), which impose $F_1\subsetneq F_2$ for two r-rooted c-forests. Without this, $S$ would always have a trivial hedge, itself, Consequently, many of our statements and proofs would need to be modified to exclude the trivial hedge. complicating many statements and proofs. For example, "hitting all of the hedges of $S$" would need to be "hitting all of the nontrivial hedges of $S$."
>
> ### Thank you for your time and detailed feedback. We are confident in the correctness of the technical statements in our paper. We appreciate your other comments, most of which are easily fixable, and will be promptly addressed in the manuscript. We kindly request that you acknowledge the soundness of our work and consider increasing your score accordingly.

---

> > ### Comment · Reviewer_tZFY · 2024-08-13
> > **Response**
> >
> > Thank you for your detailed response. Your response addresses my concerns on W3 and W4.
> >
> > ---
> >
> > > W1
> >
> > I understand that the problem of finding the minimum cost intervention set has been proposed by Akbari et al., 2022. Thank you that you appreciated my comment on W1.
> >
> > ---
> >
> > > W2
> >
> > I think this issue will not be addressed by simply changing the names and adding the citations, since results such as Proposition 1 are depending on this definition. Please check dependencies of this definition throughout the paper.
> >
> > ---
> >
> > On W6, can you provide the pointer justifying your responses? Specifically,
> >
> > > "Our reformulations not only solve the problem exactly, but also provide a significant speedup of 1,000 to 100,000,000 times for the single-district case."
> >
> > It's interesting that there is a significant speedup, given that the worst-case time complexity still remains the same. Can you discuss why this happens? Is the formal discussion included in the paper? Additionally, is there a comparison of the worst/average time complexity between the method proposed by Akbari et al., 2022, and this method? I see that there is an empirical analysis showing this speedup, but if this is a main claim of the paper, there should be a formalized discussion.
> >
> > > Moreover, for the multiple-district case, our approach is only quadratic in the number of districts (theoretically guaranteed), unlike the previous method which required iterating over an exponential number of partition sets.
> >
> > It's an interesting result. Is there a formal result justifying this claim in the paper?
> >
> > > We believe therefore the term “fast” is justified. Although the exact solution remains exponential in the worst case, our reformulations make solving these problems practically feasible and significantly more efficient, especially for large graphs.
> >
> > I think the paper contains many strong empirical results. However, to claim that the proposed method is "fast," there should be a formal discussion on how this speedup is achieved, as well as a comparison of the worst/average time complexity with previous methods.
> >
> > ---
> >
> > Overall, I agree that the paper provides an interesting approach to an important problem, as proposed by Akbari et al., 2022, within the framework of z/g-ID. Your response also provides some degree of justification for the power of the proposed method, _assuming_ your claim in the response is accurate. However, this justification is not formalized (nor included) in the paper, so as a reviewer, it's hard to take the response at face value without sufficient formalization. I believe that this paper has the potential to be strong, but it needs more work to reach perfection. Therefore, I will keep the score as is.

---

> ### Author Response · Authors · 2024-08-06
> **Limitations and minor comments**
>
> We address limitation 1 and the minor comments below.
>
>
> ## Limitations
> ### L1:
> Note that our reformulations allow for solving the MCID problem exactly. Given that the MCID problem is NP-hard, the best achievable solution for an exact algorithm is to provide a faster reformulation of another NP-hard problem, which is precisely what we have accomplished. In addition to vastly better performance in practice, our reformulations offer significant theoretical speedups for multiple-district graphs. Specifically, our algorithm operates quadratically in the number of districts, whereas the original approach is exponential. Thus, while the MCID problem is NP-hard, we have managed to solve it exactly by multiple orders of magnitude faster than the previous approach, as well as providing theoretical guarantees for a polynomial-time complexity in terms of the number of districts.
>
> ## Minor comment:
> ### W7:
> Yes, the sentence should read "... a specific class of causal effects in which the effect..." We apologize for the oversight.

---

> ### Author Response · Authors · 2024-08-13
>
> Thank you for taking the time to read our rebuttal and engage in the discussion. Please find our responses to your comments and questions below.
>
> ## W2
> Our claims, including the results we reference from other works, such as Proposition 1, are valid under our own definitions. However, it would not be difficult to use the original definitions. For example, while we did not impose maximality when defining a district, we refer to $\\{S_i\\}_{i=1}^{r}$ as the ‘maximal districts’ of $\mathcal{G}[S]$ in Proposition 1, Lemma 1, and Theorem 2, aligning with the original definition. The decision not to impose maximality in our definitions was made for simplicity, as there are sections of the paper where we discuss connected components that are not necessarily ‘maximal’, such as the set $S$ itself. Based on your suggestion, we will use standard terms for non-maximal components, but please note that our claims remain valid with this renaming.
>
> ## 1,000 to 100,000,000 speed up
> Based on the results that generated Fig. 3(a), posted below for your convenience, starting with $n$ as small as 30, we observe a speedup ratio of 1,000, which increases to 1,000,000 by $n=58$. Although we were unable to run MHS for $n$ greater than 60, we can extrapolate that for $n$ as low as 70, the speedup factor would reach at least 100,000,000.
>
> ## Worst-case complexity
> We discuss the reason behind this speed up to some extent in lines 162-173 and Remark 3 (lines 244-248). Specifically, the method of Akbari et al., 2022 has a two-step procedure:
> 1) Discover a new (minimal) hedge.
> 2) Solve a weighted minimum hitting set problem (WMHS) on the already discovered hedges.
> 3) Repeat steps 1 and 2 until the solution to the WMHS instance results in identification.
>
> We present an example in Figure 2\(c), where the number of minimal hedges is exponential in $n$. Since the WMHS problem itself is NP-hard, their approach requires running an exponential number of instances of an NP-hard problem. In comparison, our method reduces the MCID problem to a single instance of WPMAX-SAT, which has a worst-case time complexity of one WMHS problem, rather than exponentially many of them. This is why we can solve the same problem several orders of magnitude faster than the method of Akbari et al., 2022.
>
> It is noteworthy that the size of an instance of the WMHS problem they solve (i.e., the number of sets to hit) can be exponential in $n$, whereas the size of the WPMAX-SAT problem we solve (i.e. the number of variables and clauses) remains polynomial in $n$. In summary, we solve one instead of exponentially many instances of an NP-hard problem, and our problem has a significantly smaller size.
>
> We acknowledge this point isn’t clearly conveyed and will clarify it with a more detailed discussion.
>
> ## Quadratic complexity for multi-district
> We make this claim in Remark 3 of the paper. Specifically, the quadratic complexity of our method in terms of the number of districts arises from the fact that the number of variables and clauses in our SAT construction for a multi-district set $S$ with $r$ districts scales as $r^2$. The formal justification for this claim lies in Theorem 2, where we prove that MCID is equivalent to a SAT formulation with size quadratic in $r$, the number of districts.
>
> Regarding the time complexity of the approach by Akbari et al., 2022, this is due to their method requiring the enumeration of all possible partitions of the districts, the count of which (the so-called Bell number) grows super-exponentially with the number of districts. They solve an instance of single-district MCID problem for each of these possible partitions. We mentioned this in line 171.
>
> We recognize that this point may not be clearly communicated in the current manuscript, and as it is a significant contribution of our paper, we will revise the paper to ensure this aspect is highlighted more effectively.
>
> ## Unformalized justification and fast claim
> The justification for our claims is present in the paper, and we want to clarify that our response is based on information already included in the manuscript (as referenced in our responses to the formal result regarding quadratic worst-case time complexity in terms of the number of districts, discussion around how we achieve a significant speedup compared to Akbari et al., 2022, and comparison of the worst-case complexities of our method and Akbari et al., 2022). However, we acknowledge that this information may be too dispersed to effectively convey the strength of our approach. To address this, we will consolidate these justifications into a dedicated section in the revised manuscript, where we will clearly detail our contributions and demonstrate why our approach outperforms the method of Akbari et al., 2022, both theoretically and in practice.
>
> We sincerely appreciate your thorough review of our paper and your engagement with us during the discussion period. Please let us know if you have any more questions or concerns.

---

> ### Author Response · Authors · 2024-08-13
> **Summary of Fig. 3(a) as a table**
>
> ### The average time taken in seconds to solve one instance of the MCID problem for our SAT approach and the minimal hedge solver (MHS) of Akbari et al., 2022. The speedup factor refers to MHS time divided by SAT time.
> |   $n$ | SAT time               | MHS time              | Speedup factor                 |
> |:----|:-----------------------|:-----------------------|:----------------------|
> |  20 | $2.90 \times 10^{-3}$ | $1.08 \times 10^{-1}$ | $3.72 \times 10^{1}$ |
> |  31 | $1.33 \times 10^{-2}$ | $1.79 \times 10^{1}$  | $1.34 \times 10^{3}$ |
> |  42 | $3.93 \times 10^{-2}$ | $1.05 \times 10^{3}$  | $2.68 \times 10^{4}$ |
> |  53 | $8.59 \times 10^{-2}$ | $6.65 \times 10^{4}$  | $7.75 \times 10^{5}$ |
> |  58 | $1.15 \times 10^{-1}$ | $3.93 \times 10^{5}$  | $3.42 \times 10^{6}$ |
> |  61 | $1.36 \times 10^{-1}$ |         - |     - |
> |  77 | $2.95 \times 10^{-1}$ |         - |     - |
> |  86 | $4.26 \times 10^{-1}$ |         - |     - |
> |  94 | $5.70 \times 10^{-1}$ |         - |     - |
> | 100 | $7.01 \times 10^{-1}$ |         - |     - |

---

### Official Review · Reviewer_Zgff · 2024-07-10

**Soundness:** 3
**Presentation:** 2
**Contribution:** 2
**Rating:** 2
**Confidence:** 3

**Summary:**

This paper contributes to connecting the MCID problem—finding the minimum-cost interventions to identify a given causal effect, which has been proven to be NP-Complete—with four well-known problems, such as weighted maximum satisfiability and integer linear programming. These reformulations allow the original MCID problem to be solved by existing advanced solvers associated with the corresponding problems, reducing its computational complexity. The numerical experiments conducted demonstrate the improvements brought by the proposed methods.

**Strengths:**

1. The writing is based on the theoretical analysis, and the notations in the paper are well explained.
2. The topic this paper focuses on is practical, as practitioners can resort to proxy experiments when both observational studies and RCTs are difficult to collect or conduct.
3. The numerical experiments indicate the effectiveness of these reformulations and the proposed inexact algorithms.

**Weaknesses:**

1. This paper provides four types of reformulations of the MCID problem. It would be better to add a discussion section about which reformulation users should choose for their specific problem and the reasoning behind the recommendation.
2. Can the authors analyze the time complexity of the proposed heuristic algorithm?  Besides, a thorough analysis of the time complexity in experiment part would provide a deeper understanding of the algorithm's performance, particularly in terms of its behavior in worst-case and average-case conditions.
3. In lines 230-231, The inclusion of $\neg z_{k,l}$ in the SAT construction seems intuitive, but can the authors provide a more detailed explanation of its rationale? Specifically, it would be beneficial to understand how $\neg z_{k,l}$ functions within the overall logic of the problem.
4. The experimental results are solely based on synthetic data. It would be beneficial to evaluate the proposed method using real-world datasets, if possible, which would provide a more comprehensive validation of the method's effectiveness in practical applications.
5. The writing of this paper mostly focus on the theoretical results, and may be difficult to be understood by the readers, especially for the example 1, a more straightforward example may be helpful for the paper presentation.

**Questions:**

See weaknesses.

**Limitations:**

1. More details about the baselines and datasets should be provided in the manuscript. This includes a thorough description of the baseline algorithms used for comparison and the characteristics of the datasets employed in the experiments.
2. The methodology part of this paper, especially Section 3.1, might hinder readers who are not familiar with SAT from fully understanding this kind of reformulation. It would be helpful to include a more detailed and intuitive explanation of the SAT concepts and their application in this context in the appendix.

---

> ### Author Rebuttal · Authors · 2024-08-06
>
> We appreciate the reviewer’s comments and feedback. We respond to your individual comments below.
>
> ## Weaknesses:
> ### W1:
> Based on our extensive simulations given in Section 5 and Appendix A, the WPMAX-SAT reformulation, when paired with a high-performance MAX-SAT solver like RC2, consistently outperforms other approaches in terms of speed and effectiveness. The ILP reformulation, on the other hand, provides a more straightforward and interpretable way for incorporating extra constraints, say domain expert knowledge. We have included a paragraph in Section 6 emphasizing the advantages of using each reformulation.
>
> ### W2:
> Thank you for pointing this out. Indeed, the time complexity of Algorithm 1 can be analyzed in a rather straightforward way. The short answer is that the worst-case complexity of the algorithm is cubic in the number of variables. In particular, the two bottle-necks are (i) the initial pre-processing of lines 2-3, which can take up to $n$ rounds of d-separation tests (each of which can be carried out using a depth-first search), and (ii) the minimum-cut instance solved at line 9, which is carried out using a min cut - max flow algorithm such as Edmonds-Karps or linear programming, for which the worst-case complexities are known; e.g., push-relabel algorithms or MKM take time at most $O(|V|^2\sqrt{|E|})$ or $O(|V|^3)$ in the worst case. We will include the rigorous analysis in the final version.
>
> As a side note, we would like to emphasize that our Algorithm 1 serves the main purpose of solving the minimum cost generalized adjustment set, but it also can be utilized as a very perofrmant heuristic for MCID, as seen in Section 5.
>
> ### W3:
> We have provided a more detailed explanation in the text, given below:
> > We need to ensure that for each district $S_\ell$, there exists at least one intervention set $\mathcal{I_k} \in \boldsymbol{\mathcal{I}}$ such that by intervening on $\mathcal{I_k}$, $S_\ell$, becomes identifiable. However, as we do not require that *all* intervention sets hit the hedges formed for $S_\ell$, we introduce the boolean variable $z_{k,\ell}$, which indicates whether $I_k$ intersects (hits) the hedges formed for $S_\ell$, and we add the clause $(z_{1,\ell}\lor\cdots\lor z_{r,\ell})$ to the SAT expression to ensure *at least one* $z_{k,\ell}$ is $1$. Then, if $z_{k,\ell} = 1$, the solution of the $k^\text{th}$ SAT expression must hit the hedges of $S_\ell$, whereas if $𝑧_{k,\ell} = 0$, there is no such obligation. As such, the $\neg z_{k,\ell}$ term ensures that if there is no need for $\mathcal{I_k}$ to hit the hedges of $S_\ell$ (meaning that another intervention set is already hitting them), we impose no constraints on the $k$-th copy. Note that when $z_{k,\ell}=0$, or equivalently, $\neg z_{k,\ell}=1$, the $k$-th copy is already satisfied, imposing no unnecessary constraints. Note that without $𝑧_{k,\ell}$, every copy would be forced to give a solution that hits $S_\ell$, which would be undesirable.
>
> ### W4:
> Thank you for pointing this out. We have conducted new simulations on real-world networks, which we present and discuss in the general rebuttal.
>
> ### W5:
> Thank you for your valuable feedback. In response, we are revising parts of the introduction to make the concept of the MCID problem more accessible. Following your advice, we have modified the example (see the comment below) to be less technical and more straightforward.
>
> ## Limitations:
> ### L1:
> In the revised version, we are providing more detail on the baseline algorithms that we compared our methods with. Below are the relevant modified paragraphs from the main text:
>
> >	The baseline algorithm with which we compare our exact algorithms is Algorithm 2 of Akbari et al., 2022, referred to as Minimal Hedge Solver (MHS). MHS runs as follows. It maintains a growing list of discovered hedges, and at each iteration, solves a minimum hitting set problem for this list of hedges. If the solution to this minimum hitting set problem is sufficient to make the query identifiable, it terminates. Otherwise, it discovers a new hedge and grows the list of its discovered hedges. As there can be exponentially many minimal hedges, as shown in Fig. 2&#40;c&#41;, the runtime of their algorithm is doubly exponential, which is corroborated by the poor performance in our simulations.
> >
> >	We compare our heuristic algorithm, Algorithm 1, with the two best heuristics of Akbari et al., $H_1$ and $H_2$. Both algorithms solve a weighted minimum cut problem to return a set of interventions. $H_1$ performs a minimum cut between $S$ and the parents of $S$ over the bidirected edges, while $H_2$ performs a minimum cut between $S$ and the bidirected neighbors of $S$ over the directed edges.
>
> As for the datasets used, we explained them in lines 337-341 for the exact algorithms, and lines 356-359 for the heuristic algorithms. However, we have added more detail, as included below:
>
> >	For all simulations, we used the NetworkX library in Python to generate Erdos-Renyi (ER) graphs. ER graphs were generated by creating an upper triangular $n \times n$ adjacency matrix, where $n$ is the number of vertices. Then, we permuted the rows and columns of the adjacency matrix to ensure that node indices are randomized.
> >
> >	To ensure reproducibility, we used a fixed random seed for graph generation. The code as well as detailed instructions, including the specific seed values and the script for graph generation, are provided in the supplementary materials.
>
> ### L2:
> Thank you for the feedback. We will add a detailed and intuitive explanation of the WPMAX-SAT problem, starting from the SAT problem and building up to the weighted partial MAX-SAT problem. Additionally, we will include the encoding of the depth-first traversal of a directed graph as a SAT problem. This example, being similar in spirit to our approach, is easier to follow and understand, and will help readers grasp the underlying intuition behind our reformulation more effectively.

---

> ### Author Response · Authors · 2024-08-06
> **Modified example**
>
> Please find our modified example (regarding W5) below.
>
> > Consider a patient taking three types of medications: one for lowering blood pressure (antihypertensives), another for controlling diabetes (anti-diabetics), and a third for managing kidney function (renal function modulators). These medications interact in various ways to manage heart health. For example, antihypertensives directly lower blood pressure, which is closely linked to cardiovascular health. Anti-diabetics help reduce cardiovascular risk by controlling blood sugar levels. Renal function modulators can influence both blood pressure and kidney health. Lifestyle factors and other health conditions, such as metabolic syndrome, can affect how these medications work together. For instance, a healthy diet and regular exercise can enhance the effectiveness of these treatments. To understand the combined effect of these medications on heart disease risk, we could conduct a proxy experiment by adjusting the anti-diabetics, as this is often easier and safer to modify compared to the others. This approach allows us to infer how these medications collectively impact heart health.

---

> > ### Comment · Reviewer_Zgff · 2024-08-12
> >
> > Thanks for the authors' responses. I must point out that, in the rebuttal policy, it is not allowed to add an official comment before the reviewer's reply.
> >
> > Additionally, although the authors have pointed out the interesting aspects of this work, the limited evaluation regime is a disadvantage. The real-world application scenarios are semi-synthetic, which makes the results unconvincing.
> >
> > Also, I hope the authors can polish this work with more general examples so that readers can understand its challenge.
> >
> > Overall, I keep my score. Thanks for the work of the authors during the rebuttal.

---

> ### Author Response · Authors · 2024-08-13
>
> We thank the reviewer for reading and responding to our rebuttal. We address your remaining concerns below.
>
> ## Regarding limited evaluation regime
> To the best of our knowledge, no publicly available dataset includes both costs and ADMGs suitable for our setup. If the reviewer is aware of such a dataset, we would greatly appreciate the suggestion.
>
> That being said, we want to emphasize that the absence of such a dataset should not diminish the validity of our claims or the value of our contribution. First, we have **proven theoretically** that our approach operates quadratically in the number of districts of $S$, a significant improvement over the exponential complexity of the previous method (Remark 3). This theoretical advantage holds regardless of the specific dataset used.
>
> Second, our extensive simulations, detailed in Section 5 and Appendix A, involve testing over 1,000,000 graphs, covering all combinations of directed and bidirected sparsity and density, and demonstrate that our SAT reformulation outperforms the previous approach by factors ranging from 1,000 to 100,000,000 in speed. This significant speed advantage makes it highly unlikely that our approach would perform worse than the previous method in any practical scenario. Additionally, our new rebuttal simulations, which evaluated our approach on **DAGs from 17 distinct real-world applications**, demonstrate the same performance superiority. It is important to note that although the costs are synthetically generated, our experiments in Appendix A.4 confirm that varying costs does *not* impact the relative performance of our method compared to the previous approach. Therefore, our rebuttal simulations provide a reliable indication of how our approach would perform on real-world problems with non-synthetic costs.
>
> ## Regarding general examples
> We address the problem of determining optimal proxy experiments for causal effect identification, a key issue in many decision-making contexts where direct policy implementation (i.e., intervention) is costly, risky, or time-consuming. Here are two real-world examples.
>
> ### Proxy interventions for evaluating diesel vehicle bans in environmental policy​
> Consider in environmental policy, where a city is considering a ban on diesel vehicles to reduce air pollution. Instead of implementing the ban outright, the city might first introduce low-emission zones or offer incentives for electric vehicle purchases in certain areas. These low-emission zones can be viewed as proxy experiments (interventions), and they allow the city to estimate the policy's effectiveness before committing to the full-scale ban [1,2].
>
> ### Proxy experiments for evaluating nationwide sugary drink tax policies
> Consider a scenario where a government is considering a nationwide tax on sugary drinks to reduce obesity rates. Directly implementing the tax across the country could be costly and politically challenging. Instead, the government might conduct proxy experiments by imposing the tax in selected regions or by encouraging voluntary reduction in sugary drink consumption through public campaigns in others. These proxy experiments can help estimate the potential impact of a nationwide policy without the need for immediate, large-scale implementation [3,4].
>
> ## Regarding comment on rebuttal policy
> We appreciate your diligence in ensuring that the rebuttal process is followed correctly. However, we believe there might be a misunderstanding. After reviewing both the guidelines on the NeurIPS website and the emails sent by program chairs regarding the rebuttal/discussion period, we found no policy that prohibits authors from adding an official comment before the reviewer’s reply.
>
> **Please let us know if there are any remaining questions. In light of this response and our rebuttal, which addressed all of your other comments and questions, we kindly ask that you reconsider your score.**
>
> ## References
> [1]: Dey, S., Caulfield, B., & Ghosh, B. (2018). Potential health and economic benefits of banning diesel traffic in Dublin, Ireland.  _Journal of Transport & Health_.
>
> [2]: Pérez-Martínez, P., Andrade, M., & Miranda, R. (2017). Heavy truck restrictions and air quality implications in São Paulo, Brazil. _Journal of environmental management_, 202 Pt 1, 55-68.
>
> [3]: Redondo, M., Hernández-Aguado, I., & Lumbreras, B. (2018). The impact of the tax on sweetened beverages: a systematic review. _The American journal of clinical nutrition_, 108 3, 548-563.
>
> [4]: Álvarez-Sánchez, C., Contento, I., Jiménez-Aguilar, A., Koch, P., Gray, H., Guerra, L., Rivera-Dommarco, J., Uribe-Carvajal, R., & Shamah-Levy, T. (2018). Does the Mexican sugar-sweetened beverage tax have a signaling effect? ENSANUT 2016. _PLoS ONE_, 13.

---

> > ### Comment · Reviewer_Zgff · 2024-08-13
> >
> > If the official comment is rational before the reviewers' reply, What is the significance of rebuttal limiting words? Why we need to have two buttons?

---

### Official Review · Reviewer_ARx2 · 2024-07-12

**Soundness:** 3
**Presentation:** 3
**Contribution:** 3
**Rating:** 6
**Confidence:** 2

**Summary:**

In this paper, the authors consider how to introduce interventional data based on observational data to make causal effect identifiability with the minimal cost. A high-dependent method is proposed by Akbari et al. [2022], which needs a very large computational cost. In this paper, by converting the problem to weighted partially maximum satisfiability problem, the computational cost is largely reduced. The authors prove that addressing the weighted partially maximum satisfiability problem equals to find to min-cost intervention target by Theorem 1 and Theorem 2. In light of the ease to calculation by adjustment criterion, the authors also establish the relevant results for adjustment criterion.

**Strengths:**

The authors present a detailed and very clear introduction to the related studies in the literature.

The experiment results show a giant efficiency improvement compared to existing studies.

For me, it is very novel to converting the problem to a weighted partially maximum satisfiability problem, through which the problem can be addressed efficiently. And the theoretical results seem solid.

**Weaknesses:**

I do not find evident weaknesses.

**Questions:**

Could the authors provide more clues about the reason that the SAT-based method performs better than the existing method?

**Limitations:**

No.

---

> ### Author Rebuttal · Authors · 2024-08-06
>
> We thank the reviewer for the review and positive comments. We respond to your question below.
>
> ## Questions:
> ### Q1:
> Could the authors provide more clues about the reason that the SAT-based method performs better than the existing method?
>
> **Response:**
>
> At a high level, the previously existing method requires multiple calls to a subroutine solving an instance of the minimum hitting set problem, which is NP-hard. The number of calls to this subroutine can be exponential in the worst case, leading to solving exponentially many instances of a problem where each instance may require exponential time to solve. In contrast, our reformulation necessitates only a single call to a MAX-SAT solver. Additionally, the complexity of the previous methods grows exponentially with the number of districts of $S$, while the complexity of our method remains quadratic. This is due to the fact that the previous methods require enumerating all possible partitionings of the set of variables, whereas our SAT formulation avoids this unncessary enumeration by directly encoding the constraints into a single optimization problem, significantly reducing the computational burden.

---

### Official Review · Reviewer_Jqzf · 2024-07-13

**Soundness:** 3
**Presentation:** 2
**Contribution:** 3
**Rating:** 6
**Confidence:** 3

**Summary:**

The problem of finding a lower cost intervention to identify causal effects has been shown to be NP-complete.
This paper provides many new reformulations to the problem in terms of a partially weighted maximum satisfiability (in the main paper), integer linear programming (in supplementary), submodular function maximization (in supplementary), and reinforcement learning (in supplementary) that allow designing algorithms that are significantly more efficient.
The new proposed algorithms were shown to have a superior performance through extensive numerical experiments.

**Strengths:**

* The paper contributes in making the identification of causal effects with lower-cost interventions feasible in practice.

* The paper tackles a significant problem and is quite engaging; I truly enjoyed reading it.

* All proofs are provided

* All pseudo codes are provided

**Weaknesses:**

* The primary weakness of this paper lies in its presentation and clarity. I believe it would greatly benefit from another round of revision and restructuring by the authors to enhance its clarity and organization (see section questions), making it more accessible to readers who are not familiar with the topic.

* The limitations were not clearly addressed (see section limitations).

* In the introduction, the authors stated their intention to introduce several reformulations; however, only one reformulation was presented in the main paper.

**Questions:**

* Could you provide a definition of cost? Alternatively, can you offer a concrete example of what the cost might be?

* I believe the paper could be made smoother by consistently using notation. For instance, in almost all definitions, S is used for districts, but in the introduction of notations, S is used as a random set and W as a district. Please consider using S exclusively for districts. This would simplify the reading of the paper.

* lines 133 and 136 and fig 2(d): can you give an intuition of why S2 and X3 hits every hedge for S1 and S2? Maybe consider given the manipulated graph after intervention and discuss how the hedges disappear. Same remark for fig 2(b) the lines  lines 152 and 155.

* Example 1 seems to be not compatible with the text between lines 152 and 155. In the example you consider the effect of X1 and X3 on Y and you say that intervening on X2 suffices for identification; in  lines 152 and 155, you consider the effect of X2 and X3 on Y.

* Perhaps providing a brief overview of the other reformulations introduced in the supplementary material in the main paper could give readers a glimpse into the broader scope of the study. Alternatively, focusing solely on one reformulation in the main paper and leaving the others for future work might streamline the presentation.

* There is a typo in "causal effects win which the effect of ..."

**Limitations:**

Limitations were not clearly addressed in the paper, although some limitations are implicitly pointed out in the complexities and conditions under which the results hold true. It would be beneficial to explicitly discuss these limitations. Additionally, it would be valuable to explore whether there are specific limitations associated with the new reformulations of the problem compared to the initial formulation.

---

> ### Author Rebuttal · Authors · 2024-08-06
>
> We thank the reviewer for their comments and positive assessment. Taking your comments into account, we have rewritten and restructured the relevant sections of the paper to improve the presentation and clarity. We respond to your comments and questions below.
>
> ## Weaknesses:
> ### W1:
> We acknowledge the need for improvements in how the content is conveyed. We are refining our manuscript to enhance its readability based on the feedback we received. We are undertaking another round of revision and restructuring to enhance the clarity and organization.
>
> ### W2:
> Please see our answer to L1.
>
> ### W3:
> Thank you for noticing this and bringing it to our attention. Due to space limitations, we chose to present only the two most noteworthy and practical reformulations, namely WPMAX-SAT and ILP, in the main paper. We have revised the introduction to clarify that we present and focus on these two reformulations in the main text, while the other reformulations are provided in Appendix D.
>
> ## Questions:
> ### Q1:
> Costs are assigned to interventions on variables (i.e., conducting experiments). We consider heterogeneous costs, as certain experiments can be more costly to run than others. 'Cost' in this context can be in terms of:
> - **Financial Costs**: For example, in the healthcare system, different interventions can have varying financial implications. Upgrading medical equipment might be relatively inexpensive compared to implementing a comprehensive telemedicine service or enhancing emergency response systems.
> - **Time Resources**: Time is another critical aspect of cost. For instance, upgrading medical equipment in already existing hospitals may require significantly less time than building a new hospital from scratch.
> - **Other Resources**: This includes human resources and other logistical considerations. For example, implementing a new healthcare protocol might require extensive training for staff, which can vary in cost depending on the complexity and scope of the training required.
>
> Additionally, some experiments are not possible to perform as they might be unethical or have not yet recieved approval from authorities, and for these experiments, we model the cost as infinite. For example, in medical research, conducting certain experiments might require exposing patients to possibly harmful or unapproved treatments or withholding treatments currently percieved as effective, which is ethically unacceptable.
>
> We have included these concrete examples in the revised paper to help readers better understand the practical implications of costs.
>
> ### Q2:
> Thank you for pointing this out. We have adapted the notation in the introduction to be more consistent with the rest of the paper.
>
> ### Q3:
> Thank you for thoroughly going over our examples! In Fig. 2(d), notice that $\\{S_1\\}$ and $\\{S_2\\}$ are the two unique districts of $S=\\{S_1,S_2\\}$. As they are both singletons, we use $\\{S_1\\}$ and $S_1$, and $\\{S_2\\}$ and $S_2$ interchangeably for the sake of conciseness. Note that after intervening on a variable, in the resulting graph, all the incoming edges to that variable disappear, i.e., the intervened variable has no parents.
>
> Focusing first on $S_1$, we observe that after intervening on $S_2$, the resulting graph contains no hedges for $S_1$ by Definition 2. This is because the only sets $W \supsetneq \\{S_1\\}$ that are districts (i.e., connected via bidirected edges) and contain $S_1$ are $W= \\{S_1, X_2\\}$, $W= \\{S_1, X_4\\}$, or $W = \\{S_1, X_2, X_4\\}$, and neither of these sets satisfy condition (ii), namely that all $w \in W$ are ancestors of $S_1$ in $\mathcal{G}[W]$. Looking now at $S_2$, the only ancestor of $S_2$ is $X_3$, and following an intervention on $X_3$, it is not in the same district with $S_2$. Therefore, by Definition 2, there are no hedges remaining for $S_2$.
>
> We have added these explanations for both figures 2(b) and 2(d) in the revised version.
>
> ### Q4:
> You are right that there is a discrepancy between Example 1 and what we had written between lines 152 and 155. We have fixed the example to be consistent with what is stated in lines 152-155.
>
> ### Q5:
> We agree with your suggestion that focusing on one reformulation would streamline the presentation, which is why we concentrated on the WPMAX-SAT and ILP formulations. These two formulations are very similar, making it sensible to present both. To clarify this better, we have added an explanation in the main text on why we chose to focus on these reformulations:
>
> >	Based on our extensive simulations provided in Section 5 and Appendix A, the WPMAX-SAT reformulation, when paired with a high-performance MAX-SAT solver like RC2, consistently outperforms other approaches in terms of speed and effectiveness. Similarly, the ILP formulation aligns very closely with the structure of the WPMAX-SAT approach, thus justifying our focus on these two formulations.
>
> ### Q6:
> The sentence should read "... a specific class of causal effects in which the effect..." We apologize for the oversight.
>
> ## Limitations:
> ### L1:
> We have added a paragraph addressing the limitations in the main text. We provide the paragraph below:
>
> >	The main limitation of our reformulation is that it still involves solving an NP-hard problem, which inherently has a worst-case exponential complexity. However, it should be noted that the MCID problem itself is an NP-hard problem, meaning any reformulation will inevitably have an exponential worst-case complexity. Despite this, our WPMAX-SAT reformulation offers significant advantages over the initial formulation by expressing the problem as a single instance of a WPMAX-SAT problem. This approach allows for the use of well-studied and high-performance solvers, as demonstrated in our simulations. Moreover, our reformulation benefits from a quadratic complexity in the number of districts of $S$, compared to the exponential complexity of the initial formulation by Akbari et al., 2022.

---

> > ### Comment · Reviewer_Jqzf · 2024-08-11
> >
> > I thank the authors for their responses, in which they have addressed most of my concerns and have highlighted the interesting aspects of this paper. However, I also acknowledge the valid points raised by the other reviewers. Therefore, I will keep my score unchanged.

---

### Author Rebuttal · Authors · 2024-08-06

Thank you to all reviewers for your valuable feedback. We have carefully reviewed each comment and addressed all questions and concerns in our individual rebuttals. We welcome further questions or comments and look forward to engaging with you during the discussion period. We have made the following major changes and additions:

1. Streamlined our paper and clarified our focus on the WPMAX-SAT and ILP formulations, addressing concerns raised by Reviewer $\color{darkblue}{\textbf{Jqzf}}$ and Reviewer $\color{darkred}{\textbf{Zgff}}$.
2. Conducted new experiments on real-world data, as requested by Reviewer $\color{darkred}{\textbf{Zgff}}$, with results presented below.
3. Enhanced readability and ease of understanding, particularly for our SAT reformulation, in response to feedback from Reviewer $\color{darkred}{\textbf{Zgff}}$ and Reviewer $\color{darkgreen}{\textbf{tZFY}}$. We provide a walkthrough example of the SAT construction for a toy-example graph below.
4. Implemented other clarity improvements, including providing intuitions about the SAT construction and clarifying our definitions, as suggested by all reviewers.

Thank you again for your feedback. We kindly request that you reevaluate your assessment given our responses and improvements.

## New experiments on real-world data
We have conducted new experiments using 17 real-world networks from the Bayesian Network Repository [1]. This repository encompasses networks from diverse domains such as biology, engineering, medicine, and social science.

In our experiments, each network from the repository is utilized as the DAG on the known variables. We assign a random cost to each variable, sampled uniformly at random from $[1, n]$. We then introduce hidden variables by randomly sampling bidirected edges with probabilities of 0.01, 0.1, and 0.3, resulting in 50 ADMGs per network. For each ADMG, we select $S$ to be a singleton consisting of the last node in the causal order. This choice ensures that any other node would not reduce the network’s size by ignoring some vertices and edges, as outgoing edges from $S$ are irrelevant for the MCID problem.

Subsequently, we implemented the exact algorithms: minimal hedge solver (Algorithm 2 of Akbari et al., 2022), ILP, and WPMAX-SAT. We also tested the heuristic algorithms $H_1$ and $H_2$ from Akbari et al., 2022, alongside Algorithm 1 from our work.

The results are given in the uploaded PDF document. In Fig. 1, we present a semi-log bar chart of the average time taken for each exact algorithm to solve an instance of a graph for each network. Notably, our approaches, ILP and SAT, consistently outperform MHS by an average factor of over 100, and by more than 1000 times on the largest network, link. Additionally, SAT demonstrates superior performance in all but two networks, _andes_ and _diabetes_.

In Fig. 2, we provide a semi-log plot of the normalized cost of the heuristics $H_1$ and $H_2$ from Akbari et al., 2022, and our Algorithm 1, averaged across each network. It is evident that Algorithm 1 significantly outperforms $H_1$, often by one to two orders of magnitude. Although $H_2$ performs better due to its focus on performing cuts on nodes over directed edges (which are fewer), its cost remains higher than that of Algorithm 1.

Overall, the results align with our findings from synthetic simulations in the main text. Our WPMAX-SAT and ILP formulations surpass the previous state-of-the-art approach by Akbari et al., 2022, with WPMAX-SAT being the fastest. Furthermore, our polynomial-time Algorithm 1 consistently outperforms the heuristics proposed by Akbari et al., 2022.

[1] Scutari M (2010). “Learning Bayesian Networks with the bnlearn R Package.” _Journal of Statistical Software_, **35**(3), 1–22. [doi:10.18637/jss.v035.i03](https://doi.org/10.18637/jss.v035.i03).

## Walkthrough SAT construction example
Consider the graph $v_1\to v_2\to v_3$, $v_1\leftrightarrow v_2 \leftrightarrow v_3$ where $S = \\{v_3\\}$, with $n=3$ vertices. The hedge hull of $S$ is $\\{v_1,v_2, v_3\\}$, and $m=\left|\\{v_1,v_2\\}\right| =2$. Following the SAT construction procedure outlined in lines 193-203, we will carry out the construction in $m+2=4$ steps. Our SAT expression will consist of 8 variables, $x_{i,j}$ for $i\in\\{1,2\\}$ (corresponding to $v_1$ and $v_2$), and $j\in\\{0,1,2,3\\}$, corresponding to the four steps of construction.

### Step 1:
We add the clauses $(\lnot x_{1,0} \lor x_{1,1} \lor \lnot x_{2,1})$ and $(\lnot x_{2,0} \lor x_{2,1} \lor \lnot x_{3,1})$, corresponding to the edges $v_1\to v_2$ and $v_2\to v_3$, respectively. Note that by convention, $x_{3,1}=1$, and the second clause reduces to $(\lnot x_{2,0} \lor x_{2,1})$.
### Step 2:
For the edge $v_1\leftrightarrow v_2$, we add the clauses $(\lnot x_{1,1} \lor x_{1,2} \lor \lnot x_{2,2})$ and $(\lnot x_{2,1} \lor x_{2,2} \lor \lnot x_{1,2})$.
Similarly, for the edge $v_1\leftrightarrow v_3$, we add the clauses $(\lnot x_{1,1} \lor x_{1,2} \lor \lnot x_{3,2})$ and $(\lnot x_{3,1} \lor x_{3,2} \lor \lnot x_{1,2})$.
Since by convention, $x_{3,1}=x_{3,2}=1$, the latter two clauses reduce to $(\lnot x_{1,1} \lor x_{1,2})$ and $1$, respectively.
### Step 3:
We add the clauses $(\lnot x_{1,2} \lor x_{1,3} \lor \lnot x_{2,3})$ and $(\lnot x_{2,2} \lor x_{2,3} \lor \lnot x_{3,3})$, corresponding to the edges $v_1\to v_2$ and $v_2\to v_3$, respectively. Again $x_{3,3}=1$, and the latter clause reduces to $(\lnot x_{2,2} \lor x_{2,3})$.
### Step 4:
The clauses $\lnot x_{1,3}$ and $\lnot x_{2,3}$ are added.

Finally, combining all the clauses together, the SAT expression is given by
$(\lnot x_{1,0} \lor x_{1,1} \lor \lnot x_{2,1}) \land (\lnot x_{2,0} \lor x_{2,1})
\land
(\lnot x_{1,1} \lor x_{1,2} \lor \lnot x_{2,2}) \land (\lnot x_{2,1} \lor x_{2,2} \lor \lnot x_{1,2})
\land
(\lnot x_{1,1} \lor x_{1,2})
\land
(\lnot x_{1,2} \lor x_{1,3} \lor \lnot x_{2,3})
\land
(\lnot x_{2,2} \lor x_{2,3})
\land \lnot x_{1,3} \land \lnot x_{2,3}.$

---

### Decision · Program_Chairs · 2024-09-25

**Decision:**

Accept (poster)

**Comment:**

Reviewer Zgff reduced their score to a 2 from a 4 after rebuttal. Based on the discussion, it is clear they have taken this as a retaliation because the authors posted a comment since the authors answered the questions in great detail. Posting comments before reviewer engagement is not forbidden by the NeurIPS policies. I agree with the authors that this is very unprofessional/policing behavior by the reviewer, rather than evaluating the scientific merit of the work.

Because of this, I am treating Reviewer Zgff's score as a 4 instead of the updated 2. This brings the average to a 5, making this a borderline paper.

With this, overall no reviewer has a strong opinion about this paper in its current form. Two reviewers acknowledge the validity of the contributions; although admittedly one with low confidence. The critical reviewer #tZFY, who is also an expert, provided very detailed feedback. There were initial concerns around inconsistent use of some previously well-known concepts, and some around clarity. Most of these seem to have been cleared out by the rebuttal although the reviewer seems to be hesitant to increase their score without seeing a revised version with changes implemented. There seem to be no major concerns by any of the reviewers currently.